# DISTRIBUTION AWARE ACTIVE LEARNING VIA GAUSSIAN MIXTURES

## ABSTRACT

In active learning (AL), the distribution of labeled samples in a latent space is often dissimilar to that of unlabeled samples, depending on various factors such as labeled set size or data selection strategy. This distributional discrepancy hampers both evaluation and estimation of informativeness on unseen data, and remains an important issue in AL. In this paper, we propose a robust distribution-aware learning and sample selection strategy that employs Gaussian Mixture Model (GMM) to effectively encapsulate both labeled and unlabeled sets for AL. By utilizing the GMM statistics derived from all available data, the proposed approach is able to construct a more diverse feature representation, thereby reducing the risk of overfitting to limited patterns. Specifically, we propose a regularization method that supervises GMM posteriors under the concept of metric learning, and introduce a semi-supervised learning method that feeds GMM statistics into an adversarial discriminator to prevent memorization of samples. Furthermore, we propose a new informativeness metric that utilizes GMM likelihoods to detect overfitted areas in the latent space, and then devise a hybrid sample selection strategy that takes advantage of the properties of different informativeness metrics. Extensive experimental results demonstrate that our GMM-based method outperforms existing works on various balanced and imbalanced datasets, and can be readily integrated with other AL schemes to further improve the performance.

## 1 INTRODUCTION

Recent advancements in machine learning have underscored the critical demand for a substantial amount of labeled training data. However, in practical real-world applications, it is often challenging to acquire labeled data especially in cases that require specialized expertise, such as medical diagnoses or complex vision tasks. In this context, active learning (AL) emerges as a promising solution to maximize annotation efficiency, where the trained model iteratively selects the most informative subset from the unlabeled dataset to be annotated by human experts. In virtue of previous efforts, numerous AL solutions have been proposed: prediction-based methods (Wang & Shang, 2014; Joshi et al., 2009) measure the uncertainty of predictions; model-based methods (Caramalau et al., 2021; Sinha et al., 2019) train separate models to measure the informativeness of samples; feature-based methods (Sener & Savarese, 2018; Agarwal et al., 2020) utilize the relationship between feature representations on the latent space to enhance the diversity of selected samples.

However, despite these contributions, overcoming the overfitting issue still remains a fundamental challenge in AL. This issue can be observed from Fig. 1, which shows the t-SNE visualization of latent space after training the neural network with 1,000 labeled samples of CIFAR-10 after the first AL cycle. Although the model learns a semantically distinguishable representation on labeled set $X_L$, there exists a clear discrepancy between $X_L$ and unlabeled set $X_{UL}$. In other words, given a feature extractor $f_\theta$, the latent distribution becomes distorted as $p(f_\theta(X_L)) \neq p(f_\theta(X_{UL}))$. Accordingly, the model may become overfitted to this unrepresentative $X_L$, thereby compromising its ability to generalize. This discrepancy is a well-known phenomenon in AL (Farquhar et al., 2021), which can occur due to various reasons such as a small

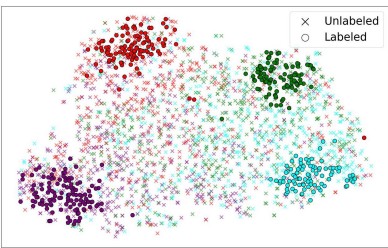

Figure 1: t-SNE visualization of latent space after $1^{st}$ AL cycle in CIFAR-10.

labeled set size during early AL cycles or a biased labeled set caused by a particular data selection strategy. It is important to mitigate the distributional discrepancy between $X_L$ and $X_{UL}$ in AL as it can deprive the task model of its generalization capability, which can be problematic when (i) evaluating the task model on test data or (ii) estimating the informativeness of unlabeled samples.

Despite its importance, only a few prior works have attempted to resolve this underlying challenge in AL. For example, (Huang et al., 2021; Gao et al., 2020; Yuan et al., 2021) propose to minimize inconsistency between predictions on unlabeled samples and use this inconsistency as informativeness metrics. (Kim et al., 2021; Su et al., 2020; Sinha et al., 2019) train discriminator modules to select samples that are expected to originate from the unlabeled set. However, performances of existing methods are potentially limited as they are not able to precisely capture the distribution statistics of $\{X_L, X_{UL}\}$, thereby failing to effectively reduce the distributional discrepancy.

**Contribution.** In this paper, we propose a new direction to AL that directly calculates the distribution statistics $\{\psi_L, \psi_{UL}\}$ of the entire dataset $\{X_L, X_{UL}\}$ and suggest (i) distribution-aware training strategies and (ii) distribution-aware informativeness metrics to mitigate the distributional discrepancy. Our key idea is to leverage a solid unsupervised clustering method, Gaussian Mixture Models (GMM), which is a useful tool to calculate statistics of the latent distribution as $\psi := (\pi, \mu, \sigma^2)$. We propose a regularization method that utilizes $\psi$ as a supervision signal to guide the learning of robust general representations against noisy samples. Additionally, we introduce semi-supervised learning methods that leverage the statistics of $\psi$ to reduce the distributional discrepancy between $X_L, X_{UL}$ and a novel informativeness metric based on the difference in GMM likelihoods. Overall, the incorporation of GMM serves as an effective approach to mitigate distortions in latent space, ultimately leading to improved performance. Our contributions are as follows:

- We propose the characterization of datasets via GMM and the incorporation of the learned GMM statistics from the entire datasets into the supervised and semi-supervised training strategies. Our method aims to learn an undistorted representation to improve the generality of the latent distribution.

- Based on the distributional knowledge from GMM, we propose a novel informativeness metric that measures the difference in likelihoods. Additionally, we introduce a hybrid selection strategy that seamlessly integrates prediction-based, model-based and likelihood-based informativeness.

We demonstrate the competence of our methods on the benchmark datasets: CIFAR-10, CIFAR-100, SVHN, FashionMNIST.Extensive experiments show that the proposed methods significantly improve performance, achieving state-of-the-art results. We also confirm their superiority on imbalanced datasets which have different numbers of samples per category.

## 2 RELATED WORKS

**Active learning.** AL aims to select a small number of informative unlabeled samples that can maximize the efficiency of the limited annotation budget. AL can be broadly categorized into two types: model-based methods and feature-based methods. First, model-based methods compute the informativeness scores of unlabeled samples based on the output of the neural network. Related informativeness metrics can be defined as the margin between the first and second confident predictions (Joshi et al., 2009; Roth & Small, 2006); entropy of the predictive distribution (Wang & Shang, 2014; Settles & Craven, 2008; Luo et al., 2013); variance in the model ensembles (Beluch et al., 2018; Haussmann et al., 2020; Jung et al., 2022). Recently, there have been significant efforts to train a separate module that predicts only the informativenss of samples. For example, (Sinha et al., 2019; Kim et al., 2021) measure the unfamiliarity of data samples through adversarial training, while (Yoo & Kweon, 2019) trains a loss prediction module to approximate the informativeness score. However, these model-based methods are known to be vulnerable to redundant sampling where similar samples are chosen at the same time, decreasing total informativeness of the selected subset. Second, feature-based methods use the relationship between feature representations on the latent space to measure the informativeness score. For instance, (Sener & Savarese, 2018; Agarwal et al., 2020) solve the K-center problem to enhance the diversity of a core-set, and (Parvaneh et al., 2022) uses interpolation to detect informative areas in latent space, while (Caramalau et al., 2021) measures the distance between features via a graph neural network. However, these existing feature-based methods have primarily focused on selecting diverse instances; these approaches are vulnerable to outlier samples and may overlook specific regions that intensively contain informative samples, potentially

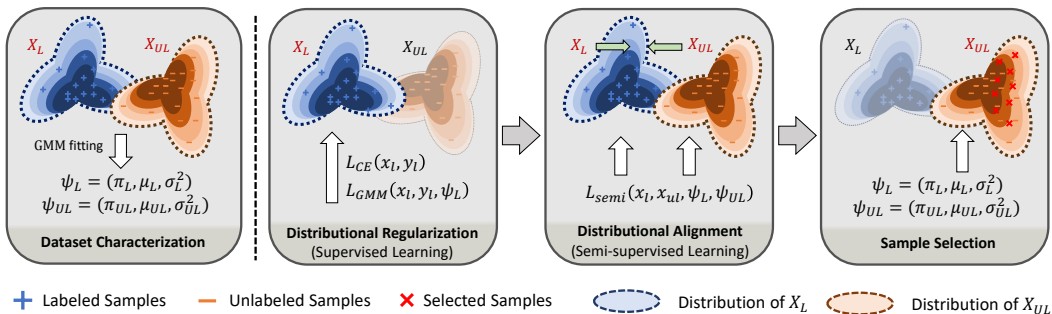

+ Labeled Samples    − Unlabeled Samples    ✗ Selected Samples    Distribution of $X_L$    Distribution of $X_{UL}$

Figure 2: **An overview of the proposed AL framework**. Our primary idea is to extract distributional statistics from $X_L, X_{UL}$ via GMM. This learned GMM statistics $\psi_L, \psi_{UL}$ is used to train the task model and to select informative samples. At every AL learning cycle, 1) the model is first trained during the supervised learning stage with the aid of auxiliary distribution-aware regularizer $L_{GMM}$. 2) Next, during the semi-supervised learning stage, latent distributions of $X_L, X_{UL}$ are aligned to have small Wasserstein distance using $\psi_L, \psi_{UL}$. 3) At the selection stage, the informativeness of unlabeled samples is measured through $\psi_L, \psi_{UL}$ for annotation.

leading to the reduction of resultant performance. In contrast, our work goes beyond merely increasing diversity. We directly calculate the distributional statistics $\psi$ of datasets $\{X_L, X_{UL}\}$ via GMM, and selectively identify informative samples that minimize the distributional discrepancy between $X_L$ and $X_{UL}$ to effectively mitigate overfitting and improve model performance.

**Semi-supervised active learning.** Early approaches to semi-supervised AL primarily treated the task model as a black box and focused on leveraging unlabeled data solely for sample selection, rather than utilizing it to directly train the task model. For example, VAAL (Sinha et al., 2019) and TA-VAAL (Kim et al., 2021) trained a separate variational autoencoder exclusively for selection, while CoreGCN (Caramalau et al., 2021) similarly trained a graph convolutional network as a sampler. Although these methods employed unlabeled data to train auxiliary modules, their purpose was for sample selection, without making any direct impact on the performance of the task model. In contrast, recent methodologies have emerged that make use of unlabeled data to train the task model to further enhance the performance of the task model. (Siméoni et al., 2021) pioneered this field by exploiting pseudo labels obtained from label propagation, and TOD (Huang et al., 2021) proposed minimizing temporal output discrepancy in unlabeled samples. WAAL (Shui et al., 2020) presented theoretical insights based on distribution matching that unify model training and sample selection. Compared to previous semi-supervised active learning works, our proposed approach takes a direct approach by extracting distributional statistics from entire datasets via GMM. This allows us to construct an efficient distribution-aware semi-supervised learning framework that uses the global distributional information to mitigate the risk of overfitting the task model to scarce labeled samples, thereby improving generalization capability of the task model in a low-data setting.

## 3 PROPOSED METHOD

### 3.1 PROBLEM SETUP AND ORGANIZATION

Given unlabeled total dataset $X$, AL framework randomly samples and annotates the initial labeled set $X_L^0$ from $X$ while the rest remains unlabeled as $X_{UL}^0 = X / X_L^0$. Note that the size of the initial labeled set $X_L^0$ is much smaller than that of $X_{UL}^0$ (i.e., $|X_L^0| << |X_{UL}^0|$). In AL, DNN is repeatedly initialized and trained during successive active learning cycles $t = 0, 1, .., T$. At the end of each $t^{th}$ cycle, the DNN selects a small subset of unlabeled samples $S^t$ with the highest informativeness. Then, human experts annotate $S^t$ to update the labeled set as $X_L^{t+1} = X_L^t \cup S^t$ and the unlabeled set as $X_{UL}^{t+1} = X_{UL}^t \setminus S^t$. The main goal of this work is to develop a learning and data selection strategy that can effectively handle the overfitting problem of AL illustrated in Fig. 1.

**Overview of our method.** Fig. 2 illustrates the overview of our approach. Our core idea is to optimize GMM parameters $\{\psi_L, \psi_{UL}\}$ to $\{X_L, X_{UL}\}$ for the probabilistic characterization of the global data distribution (Section 3.2). The acquired data statistics $\{\psi_L, \psi_{UL}\}$ are then used to improve training and selection stages in our framework. First, in the supervised-learning stage (Section 3.3), the task model is trained with the proposed regularization method that relocates samples to be consistent with $\{\psi_L, \psi_{UL}\}$. Second, in the semi-supervised learning stage (Section 3.4), we propose

Figure 3: During supervised learning, posterior probabilities $p(y = k|z; \psi_L)$ for each Gaussian component are computed using the learned GMM parameters $\psi_L$. Whereas $L_{CE}$ deals with logits of the classifier, $L_{GMM}$ focuses on the latent space relocating samples closer to the correct modality, yet farther away from the others.

to feed $\{\psi_L, \psi_{UL}\}$ into a discriminator module to effectively minimizes the Wasserstein distance between $X_L$ and $X_{UL}$. Lastly, in the selection stage (Section 3.5), we propose a novel distribution-aware informativeness metric that compares GMM likelihoods of samples to detect overfitted latent areas, as well as a hybrid selection method that is designed to choose from uncertain yet diverse unlabeled samples. Note that $\{\psi_L, \psi_{UL}\}$ are updated repeatedly to reflect the change in latent spaces. In the following sections, we present our distribution-aware training and selection methods step by step.

## 3.2 CHARACTERIZATION OF DATASETS VIA GMM

We probabilistically characterize the feature representation of the training data using isotropic Gaussian Mixture Models (GMM). The number of Gaussian modality in GMM is set to be matched to each category of the training data. This characterization can be viewed as a generalization of k-means clustering, taking into account mixing coefficients and covariance structures simultaneously. Given samples $\{x_i\}_{i=1}^N$ and their feature representations $\{z_i = f_\theta(x_i)\}_{i=1}^N$, we optimize the parameter set $\psi := (\pi, \mu, \sigma^2)$ of GMM using the Expectation Maximization (EM) algorithm:

(**E Step**) Compute the responsibility $\gamma$ from the current parameter set $\psi := (\pi, \mu, \sigma^2)$

$$\gamma_{ik} := p(y_i = k|z_i) = \frac{\pi_k \mathcal{N}(z_i|\mu_k, \sigma_k^2)}{\sum_{j=1}^K \pi_j \mathcal{N}(z_i|\mu_j, \sigma_j^2)}$$

(**M Step**) Update the parameter set $\psi := (\pi, \mu, \sigma^2)$ based on the current responsibilities

$$\pi_k = \frac{\sum_{i=1}^N \gamma_{ik}}{\sum_{k=1}^K \sum_{i=1}^N \gamma_{ik}}, \quad \mu_k = \frac{\sum_{i=1}^N \gamma_{ik} z_i}{\sum_{i=1}^N \gamma_{ik}}, \quad \sigma_k^2 = \frac{\sum_{i=1}^N \gamma_{ik}(z_i - \mu_k)^2}{\sum_{i=1}^N \gamma_{ik}}$$

where $\pi_k, \mu_k, \sigma_k$ respectively denote mixing coefficient, mean, and diagonal covariance of $k^{th}$ modality while $K$ is the number of categories. Note that $\gamma_{ik}$ stands for the estimated 'responsibility' of $k^{th}$ Gaussian modality $\mathcal{N}(z|\mu_k, \sigma_k^2)$ for generating the data $z_i$. When dealing with labeled samples, we fix the values of $\gamma_{ik}$ to be binary (e.g., 0 or 1), since there is no need to estimate categories of labeled samples. To handle the memory constraints arising from the large size of $X_L$ and $X_{UL}$, we divide datasets into multiple batches and fit a GMM to each batch; afterwards, we compute the average of learned GMM parameters $\psi$ obtained from all batches, treating this average as probabilistic representations of the datasets, denoted as $\{\psi_L, \psi_{UL}\}$ for $\{X_L, X_{UL}\}$ respectively. For unlabeled samples, it is noteworthy that the order of clusters can be scrambled at every batch. Thus, we solve a Hungarian matching problem to realign the order of each cluster consistently. Regarding the cost function for the alignment, we compute Bhattacharyya distance between each Gaussian modality. Bhattacharya distance between two multivariate normal distribution $p_k = N(\mu_k, \Sigma_k)$ is simplified as:

$$D_B(p_1, p_2) = \frac{1}{8}(\mu_1 - \mu_2)^T \Sigma^{-1}(\mu_1 - \mu_2) + \frac{1}{2} \ln(\frac{\det \Sigma}{\sqrt{\det \Sigma_1 \det \Sigma_2}}) \tag{2}$$

where $\Sigma = (\Sigma_1 + \Sigma_2)/2$. As illustrated in Fig. 2, this underlying distributional information provides essential insights during the subsequent training and selection stages of the proposed algorithm.

## 3.3 DISTRIBUTIONAL REGULARIZATION

Based on the GMM statistics obtained above, we propose the first component of our solution to address the overfitting issue. We utilize the concept of metric learning (Kaya & Bilge, 2019) and

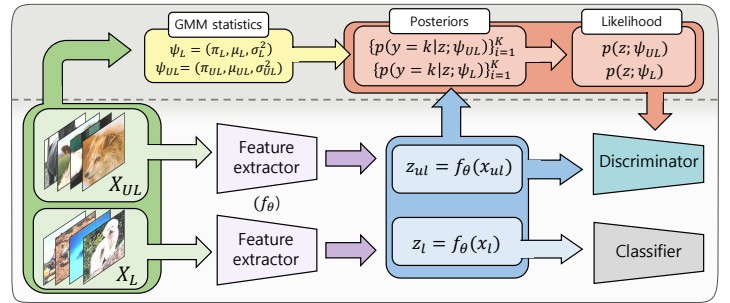

Figure 4: Proposed distribution-aware domain alignment method.

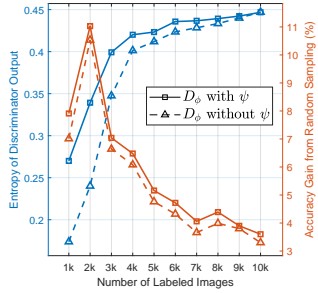

Figure 5: Analysis of discriminators on CIFAR-10.

introduce a distance-based regularization method to learn a more generalized representation where similar instances are closer together while dissimilar instances are farther apart. Fig. 3 provides a high-level description of our method within a supervised learning framework, where we propose a distribution-aware regularization loss $L_{GMM}$ that utilizes GMM statistics $\psi_L$. This loss is defined as the negative log-likelihood loss of GMM posterior: $L_{GMM}(z, y; \psi_L) = \sum_{k=1}^{K} -y_k \log(p(y = k|z; \psi_L))$, where $k$ indexes Gaussian component in equation 1. It is important to note that $p(y = k|z; \psi_L)$ denotes the probability that $z$ was generated by the $k^{th}$ Gaussian component in the GMM. As such, it serves as a soft measure of distance that captures the proximity of data point $z$ to the estimated Gaussian components $N(\mu_k, \sigma_k^2)$. Consequently, the training objective for the supervised learning stage is modified to $L_{CE}(z, y) + \alpha L_{GMM}(z, y; \psi_L)$, where $\alpha$ is a constant weight.

By incorporating $L_{GMM}$ as a regularizer, we are able to mitigate the impact of noisy or outlier data points by encouraging consistent similarity relationships rather than focusing on absolute values. Additionally, supervision using the global data distribution $\psi$ enhances robustness against noisy local batches, which is especially effective when dealing with imbalanced datasets.

### 3.4 DISTRIBUTION-AWARE ALIGNMENT WITH GMM STATISTICS

**Distributional alignment with adversarial learning.** After the supervised learning stage in Section 3.3 is completed, the semi-supervised learning stage begins subsequently. In this subsection, we introduce a distribution-aware semi-supervised learning strategy that utilizes GMM statistics to mitigate the overfitting. Our strategy is based on the theoretical analysis of (Shui et al., 2020; Mahmood et al., 2021) that the expected risk in AL is upper-bounded by a distance between the distributions of labeled and original dataset. We consider minimizing the Wasserstein distance between $X_L$ and $X_{UL}$, $W(z_L, z_{UL}) = \inf_{\delta \in \Pi(z_L, z_{UL})} \mathbb{E}_{(z_l, z_{ul}) \sim \delta}[||z_l - z_{ul}||]$, since other popular metrics (e.g., KLD, KSD) are not tractable for GMMs as they do not exist in closed-forms. On the basis of Kantorovich-Rubinstein Duality Theorem, this objective can be transformed into an adversarial learning form, as done in discriminator-based AL methods (Kim et al., 2021; Shui et al., 2020):

$$\min_\theta \max_\phi \mathbb{E}_{z_{ul} \sim f_\theta(X_{UL})}[D_\phi(z_{ul})] - \mathbb{E}_{z_l \sim f_\theta(X_L)}[D_\phi(z_l)] \qquad (3)$$

where $f_\theta$ is the feature extractor and $D_\phi$ is the 1-Lipschitz discriminator. The above training objective guides $D_\phi$ to output $D_\phi(z_l) \to 0$, $D_\phi(z_{ul}) \to 1$, while $f_\theta$ is trained to confound $D_\phi$. Additionally, we minimize the CE loss using labeled data to ensure that the adversarial learning process in equation 3 does not excessively compromise the performance of the main task. Note that we employ gradient penalty of (Gulrajani et al., 2017) to enforce the 1-Lipshitz continuous constraints on $D_\phi$.

**Distribution-aware alignment with GMM statistics.** Previous discriminator-based AL methods (Kim et al., 2021; Shui et al., 2020) often feed only the latent features $z$ of samples into the discriminator $D_\phi$. However, this architecture often causes $D_\phi$ to memorize a few labeled samples themselves, instead of capturing the overall latent structure comprehensively. In this context, as illustrated in Fig. 4, we propose to feed $D_\phi$ with distributional information from $\{\psi_L, \psi_{UL}\}$ on top of $z$, to help $D_\phi$ aware the global data distribution. Given that $\psi$ is a group of high-dimensional vectors (e.g., $\mu_L \in R^{100 \times 512}$ for CIFAR-100), it is hard to simply concatenate $\{\psi_L, \psi_{UL}\}$ to $z$, so we instead propose to indirectly exploit $\{\psi_L, \psi_{UL}\}$ for the sensible implementation. Specifically, we first compute the posterior probabilities $\{p(y = k|z; \psi_L)\}_{k=1}^K$, $\{p(y = k|z; \psi_{UL})\}_{k=1}^K$

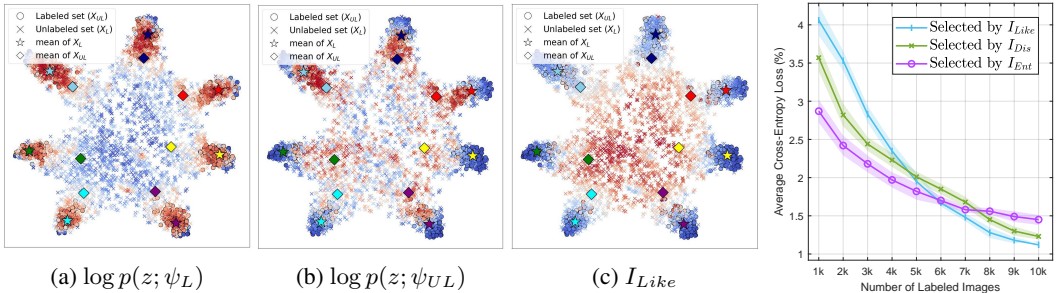

Figure 6: t-SNE of latent spaces where colors of markers represent (a,b) GMM likelihoods or (c) the proposed metric. Warmer colors indicate higher values.

Figure 7: Loss ranges of selected samples.

and the likelihoods $p(z; \psi_L)$, $p(z; \psi_{UL})$, where the likelihood $p(z; \psi)$ reflects the probability that data $z$ originates from a GMM parameterized by $\psi$. We then concatenate logarithms of the obtained probabilities to $z$ and pass the concatenated vectors to the discriminator.

To gain insights into the effect of distributional information $\psi$, in Fig. 5, we compare the average entropy values of $D_\phi$ outputs. The discriminator $D_\phi$ performs binary classification and lower entropy corresponds to more confident predictions. It is observed that vanilla $D_\phi$ without $\psi$ manifests considerably lower entropies in early cycles, which means that outputs consist of almost either 0 or 1 due to overconfident predictions. On the contrary, utilization of $\psi$ mitigates overfitting of the discriminator by considering global distribution and improves performance and entropy of prediction.

## 3.5 Distribution-Aware Informative Sample Selection

In previous subsections, we described our distribution-aware model training methods designed to reduce overfitting. In this subsection, we elaborate on our sample selection scheme for AL that aims to mitigate overfitting by reducing distributional discrepancy between $X_L$ and $X_{UL}$.

**Likelihood-based informativeness metric.** We propose a new informativeness metric that alleviates the distributional discrepancy between $\{X_L, X_{UL}\}$ by comparing GMM likelihoods using $\{\psi_L, \psi_{UL}\}$. Our metric is based on the insight that the latent space densely populated by labeled samples is already been learned, resulting in lower importance; conversely, regions with a high concentration of unlabeled samples represent unexplored areas with higher importance. Consequently, our metric aims to identify samples that not only best represent $X_{UL}$ but also differ from $X_L$, such that the distributional discrepancy can be effectively reduced. In this context, one natural question is on how we can assess the informativeness of unlabeled sample $z_{ul}$ for $\{X_L, X_{UL}\}$. To this end, we utilize the GMM likelihood $p(z_{ul}; \psi) = \sum_{j=1}^{K} \pi_j \mathcal{N}(z|\mu_j, \sigma_j^2)$ to measure how well $z_{ul}$ represents the dataset that is parameterized by $\psi$. Accordingly, we propose the likelihood-based metric $I_{Like}$ to prioritize samples that best represent unfamiliar $X_{UL}$ while being different from familiar $X_L$ as:

$$I_{Like}(x_{ul}; \psi_L, \psi_{UL}) = \log p(z_{ul}; \psi_{UL}) - \log p(z_{ul}; \psi_L), \quad p(z; \psi) = \sum_{j=1}^{K} \pi_j \mathcal{N}(z|\mu_j, \sigma_j^2) \quad (4)$$

Our intuition can be visually confirmed in Fig. 6, where we illustrate the log-likelihood $\log p(z; \psi)$ of data features $z$ from the viewpoints of $\{X_L, X_{UL}\}$ using the GMM parameters $\{\psi_L, \psi_{UL}\}$. In Fig. 6a, it can be observed that latent distribution of $X_L$ is concentrated around limited areas (indicated by hot colors), while that of $X_{UL}$ in Fig. 6b is scattered across fairly wide different areas; we conjecture that this distortion in the latent space is a result of overfitting of the model to $X_L$. In Fig. 6c, $I_{Like}$ highlights regions with a higher concentration of unlabeled samples, but fewer labeled samples. As a result, if a data sample is located in an area where $p(z_{ul}; \psi_{UL})$ is high while $p(z_{ul}; \psi_L)$ is low simultaneously, it will be selected for annotation (e.g., samples around the green, yellow markers). Conversely, even if $p(z_{ul}; \psi_{UL})$ is high, samples with high $p(z_{ul}; \psi_L)$ will be ignored due to their lower information gain (e.g., samples around the navy, cyan markers).

**Hybrid acquisition function.** While $I_{Like}$ effectively detects informative latent areas by comparing the likelihoods, it can be susceptible to redundant sampling as observed in Fig. 8a. To address the redundancy issue, we diversify the source of informativeness by utilizing the output of different models. First, we incorporate outputs of the discriminator $D_\phi$ that is learned during the semi-supervised

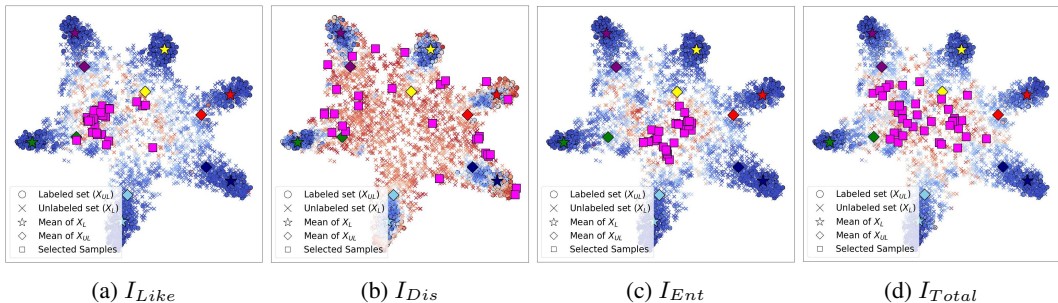

Figure 8: t-SNE of latent spaces where colors of markers represent (a,b,c) three different informativeness metics and (d) the final aggregated metric (warmer colors indicate higher values). Each metric has distinct yet complementary properties, and the aggregated metric $I_{Total}$ can select informative yet non-redundant samples.

learning, denoted as $I_{Dis} = D_\phi(z)$. Similar to $I_{Like}$, $I_{Dis}$ prioritizes the samples that are expected to be from $X_{UL}$ but relies on the output of $D_\phi$ instead of $\psi$. Fig. 8b demonstrates that while $I_{Dis}$ evenly highlights regions where labeled samples do not exist, it lacks the ability to specify particularly informative regions. Secondly, we leverage the Shannon entropy $I_{Ent}$ computed from the final logits. As shown in Fig. 8c, it provides a distinct type of informativeness that differs from both $I_{Like}$ and $I_{Dis}$. Finally, due to the different ranges and distribution shapes of the three information metrics ($I_{Like}$, $I_{Dis}$, $I_{Ent}$), we apply rank-normalization operations to each metric and then combine them to obtain the final acquisition function for selection as:

$$I_{Total}(x_{ul}) = Rank(I_{Like}(x_{ul}; \psi_L, \psi_{UL})) + Rank(I_{Dis}(x_{ul})) + Rank(I_{Ent}(x_{ul})). \quad (5)$$

Note that none of the three metrics utilize label information. As observed in Fig. 8d, incorporating information from multiple sources helps detect informative region that is both non-redundant and compensatory. Furthermore, the analysis of loss ranges in Fig. 7 provides additional justification for the aggregation of three metrics. In Fig. 7, we report averages of cross-entropy losses of 5K samples selected by each metric, confirming that the importance of each metric changes throughout the learning cycle. For example, $I_{Like}$ detects the most difficult samples in the early AL cycles, due to its ability to identify overfitted regions. On the other hand, $I_{Ent}$ becomes more effective at finding difficult samples in late AL cycles, as the distributions of $X_L$ and $X_{UL}$ become similar. After the computation of $I_{Total}$, the candidate set $S$ consisting of samples with the highest $I_{Total}$ is sent to the oracle for annotation. To mitigate redundancy within $S$, we ensure a class-wise balance using pseudo-labels and employ K-means clustering to enhance its diversity, following the methodology outlined in (Parvaneh et al., 2022; Ash et al., 2019).

## 4 EXPERIMENTS

### 4.1 EXPERIMENTAL SETUP

**Dataset.** Our proposed work is evaluated on following popular benchmark datasets: SVHN (Netzer et al., 2011), Fashion-MNIST (Xiao et al., 2017), CIFAR-10 (Krizhevsky et al., 2009) and CIFAR-100 (Krizhevsky et al., 2009). For all datasets except CIFAR-100, 1,000 samples are initially labeled and additionally 1,000 samples are labeled at the end of every cycle until the size of labeled set reaches 10,000. As for CIFAR-100, size of the $X_L$ increases from 2,000 to 20,000 in steps of 2,000.

**Implementation details.** Following the experimental setup of (Kim et al., 2021; Caramalau et al., 2021), we implement the main classifier using ResNet-18 (He et al., 2016) which is combined with a single linear layer softmax classifier. The classifier is optimized via SGD optimizer with a learning rate of 0.1; a momentum of 0.9; a batch size of 100; an epoch number of 200. The discriminator is composed of three linear layers with a sigmoid activation and optimized by Adam optimizer for 10,000 iterations with a learning rate of 5e-4 which is decayed to 5e-5 for the last 2,000 iterations. During the supervised learning stage in Section 3.3, $\psi_L$ is newly computed at the every batch since latent space keeps being updated as DNN is trained. To relieve computational burden, a balanced subset $X_s$ is sampled from $X_L$ at the every batch and GMM is fitted to $X_s$ to get $\psi_s$; this $\psi_s$ replaces $\psi_L$ in $L_{GMM}$. During the distributional alignment in Section 3.4, we alternatively update the discriminator $D_\phi$ and the remaining models in an adversarial manner for 10,000 iterations; GMM statistics ($\psi_L$, $\psi_{UL}$) are obtained at the every 500 out of 10,000 iterations. The number of iteration in EM algorithm is 10, and we cut the gradient graph to $\psi$ to reduce memory/computation complexity.

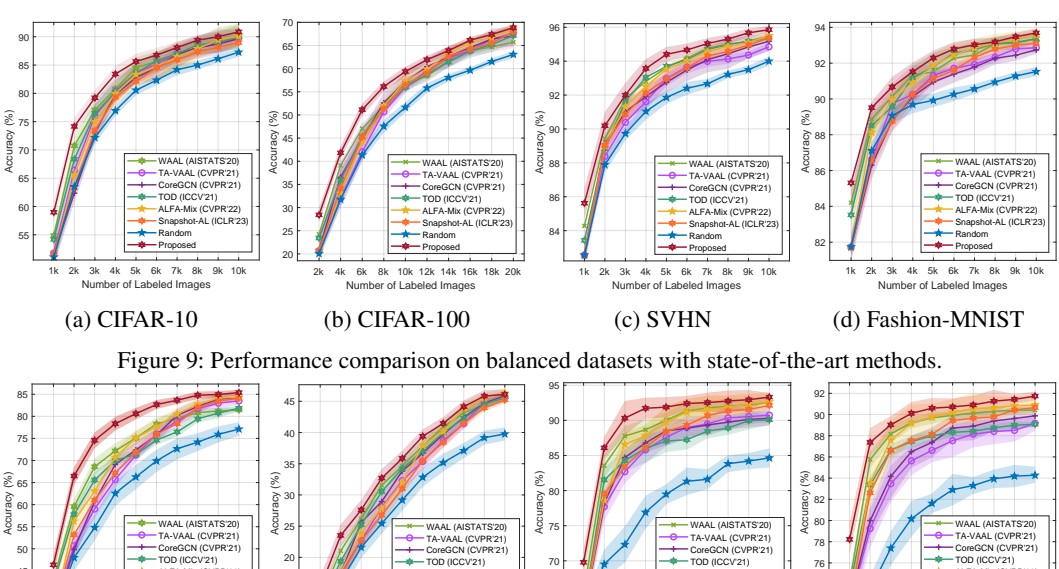

Figure 9: Performance comparison on balanced datasets with state-of-the-art methods.

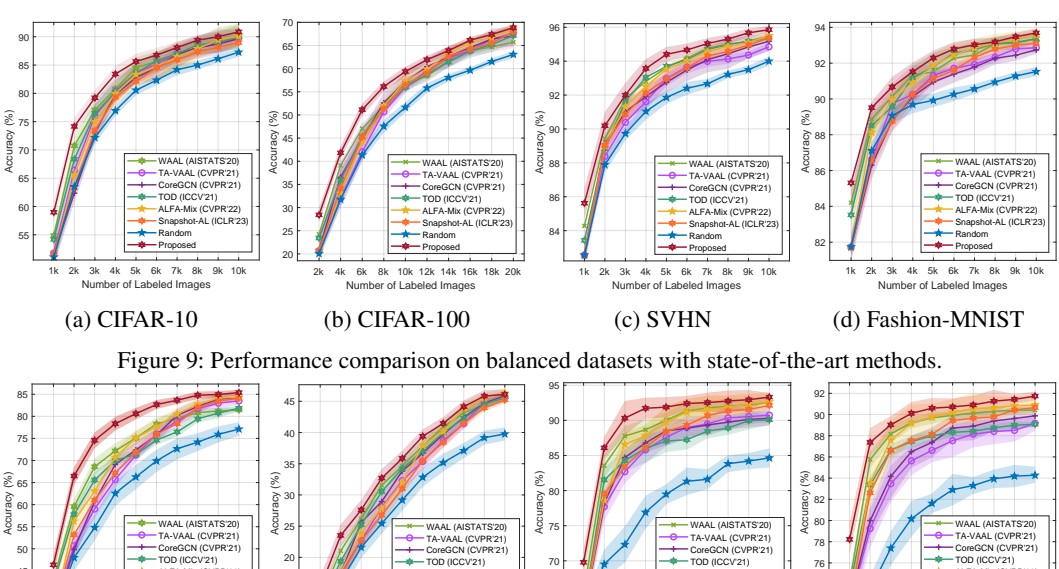

Figure 10: Performance comparison on imbalanced datasets with state-of-the-art methods.

**Baselines.** We compare our work with the state-of-the-art AL baselines: TA-VAAL (Kim et al., 2021), WAAL (Shui et al., 2020), TOD (Huang et al., 2021), ALFA-MIX (Parvaneh et al., 2022), Snapshot-AL (Jung et al., 2022) and CoreGCN (Caramalau et al., 2021), where WAAL and TOD specifically adopt semi-supervised learning to train the task model. For fair comparisons, we train five independent networks with different seeds and report the average performance with 95% confidence interval. Additional comparisons with CoreSet (Sener & Savarese, 2018), LL4AL (Yoo & Kweon, 2019), VAAL (Sinha et al., 2019) can be found in Supplementary Material.

## 4.2 EXPERIMENTAL RESULTS

**Results on balanced dataset.** In Fig. 9, we compare the performance of our method on balanced datasets with various baselines. As seen, the proposed method performs the best on all datasets, demonstrating the general applicability. In particular, the result on CIFAR100 indicates that the proposed algorithm can capture distinct Gaussian modalities, even in datasets with a large number of classes. Also, it is noticeable that our method exhibits a particular excellence in early AL cycles, performing even better than existing semi-supervised AL methods, since the proposed regularization and distributional alignment components effectively mitigate the overfitting to the small labeled set.

**Results on imbalanced dataset.** In Fig. 10, we consider a class-imbalance scenario where an imbalance ratio is set to 10 (i.e., half of all classes have 10 times more samples than those of remaining half classes). Fig. 10 shows that the proposed approach outperforms other baselines and exhibits significantly larger performance gaps when compared to the results observed in the class-balance scenario depicted in Fig. 9. To investigate the rationale behind this performance gain, we provide some analyses in Fig. 11. First, Fig. 11a depicts the t-SNE of latent space from imbalanced CIFAR-10 that all modalities (including the minor classes) are well-separated at significant distances. The reason behind this is that employing distribution-aware training, which utilizes global data statistics $\psi$ obtained from all available data, effectively prevents the model from overfitting to the biased knowledge present within the imbalanced minibatch. Additionally, the distribution-based informativeness metric guides the selection of unbiased samples. Second, Fig. 11b reports the entropy of category ratios within the labeled set. It is worth noting that a higher entropy indicates better balance among the labeled samples, with the *random* selection method exhibiting the lowest entropy (i.e., highest imbalance). In early cycles, it can be seen that all baselines prioritize samples from minor classes, leading to an increase in entropy. However, as a sufficient number of minor samples are secured in late cycles, the entropy decreases again since informative samples from major classes are

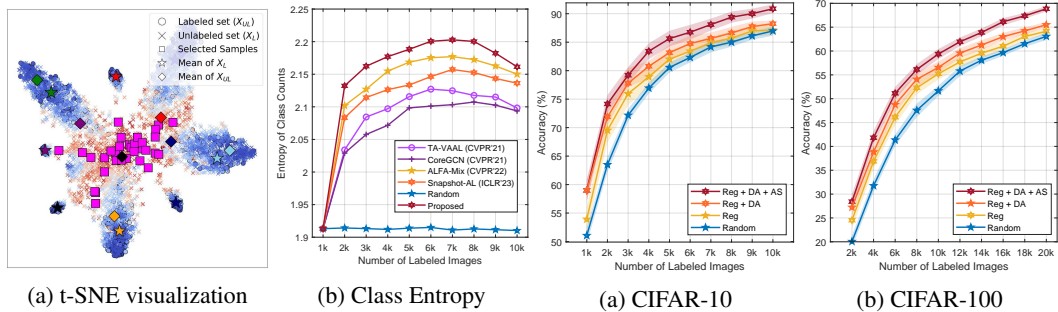

(a) t-SNE visualization    (b) Class Entropy

Figure 11: Analysis in imbalanced CIFAR-10.

(a) CIFAR-10    (b) CIFAR-100

Figure 12: Ablation studies on proposed components.

| Methods | Accuracy (%) on Proportion (%) of Labeled Samples | | | | | | | | | | |
| --- | --- | --- | --- | --- | --- | --- | --- | --- | --- | --- | --- |
| | 2.0 | 4.0 | 6.0 | 8.0 | 10.0 | 12.0 | 14.0 | 16.0 | 18.0 | 20.0 | Avg |
| ALFA-MIX (Parvaneh et al., 2022) | 35.50 | 56.25 | 63.06 | 70.71 | 75.11 | 77.63 | 80.51 | 82.72 | 84.13 | 84.35 | 71.00 |
| Reg (+ ALFA-MIX) | 42.31 | 60.92 | 66.02 | 71.75 | 76.27 | 78.50 | 81.79 | 82.92 | 84.23 | 84.53 | 72.92 |
| Reg + DA (+ ALFA-MIX) | **46.37** | **62.02** | **71.25** | **75.90** | **78.88** | **81.50** | **82.88** | **83.63** | **84.33** | **84.88** | **75.16** |
| TA-VAAL (Kim et al., 2021) | 35.73 | 50.68 | 59.02 | 65.69 | 71.23 | 75.98 | 78.89 | 81.73 | 82.98 | 83.49 | 68.54 |
| Reg (+ TA-VAAL) | 40.46 | 55.26 | 63.20 | 68.11 | 74.72 | 77.83 | 79.51 | 82.10 | 83.51 | 83.92 | 70.86 |
| Reg + DA (+ TA-VAAL) | **46.37** | **59.10** | **68.11** | **72.16** | **78.46** | **80.61** | **81.96** | **83.13** | **83.94** | **84.13** | **73.50** |
| Proposed | **46.37** | **66.48** | **74.57** | **78.33** | **80.61** | **82.67** | **83.64** | **84.78** | **84.95** | **85.38** | **76.78** |

Table 1: Compatibility analysis of our regularization (Reg) and distributional alignment (DA) strategies with other AL methods using imbalanced CIFAR-10.

preferred. Throughout each cycle, our proposed approach consistently achieves higher entropy values by selecting a significant number of minor samples. This observation highlights how leveraging distributional information $\psi$ guides the model away from bias and helps select balanced samples.

**Ablation studies on proposed components.** Fig. 12 shows ablation studies conducted on our proposed methods. For brevity, we abbreviate the distribution-aware regularization in Section 3.3 as *Reg*; distribution alignment in Section 3.4 as *DA*; active sample selection in Section 3.5 as *AS*. In every cycle, the results consistently demonstrate the gradual improvement in performance achieved by the proposed methods. Notably, *Reg* and *DA* significantly contribute to the performance gain in early cycles, alleviating overfitting to the smaller labeled sets. Conversely, *AS* steadily enhances performance even in later cycles by employing the hybrid informativeness metric for selection.

**Compatibility with other AL methods.** Our proposed learning strategies (*Reg*, *DA*) can be easily combined with existing AL methods. In Table 1, we incorporate *Reg* and *DA* into the learning stages of ALFA-MIX (Parvaneh et al., 2022) and TA-VAAL (Kim et al., 2021) while keeping the selection stages unchanged. Similar to Fig. 12, it can be seen that proposed distribution-aware learning methods improve performance by mitigating the overfitting inherent in existing AL works, and this phenomenon is especially noticeable in early cycles where the size of the labeled set is small. Furthermore, it is observed that the performance of the proposed methods (shown in the lowest row) surpasses that of previous works combined with *Reg* and *DA*. This observation suggests that our distribution-aware hybrid selection methods are better suited for choosing informative subsets, leading to enhanced overall performance.

## 5    CONCLUSION

In this study, we introduced a unified framework for active learning that leverages Gaussian Mixture Models (GMMs). Our approach involves fitting GMMs to both the labeled and unlabeled sets, effectively extracting distributional information. This information is utilized in various ways within the active learning process, including distribution-aware regularization, distributional alignment, and informativeness metrics. Through extensive comparisons with baselines and ablation studies, we demonstrate the superiority of our proposed methods. In the future, we plan to extend our research to diverse computer vision tasks such as object detection (Haussmann et al., 2020), as well as different settings like open set (Park et al., 2022) and model evaluation (Kossen et al., 2022).

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

# A APPENDIX

## A.1 ADDITIONAL EXPERIMENTAL SETUP

In this section, we describe various details involved in implementing the proposed algorithm:

- Data preprocessing followed the setting of TA-VAAL (Kim et al., 2021), that is only composed of input normalization and random augmentation with horizontal flipping and cropping.

- The task model utilized ResNet18, as commonly employed in previous works (He et al., 2016; Kim et al., 2021). It consisted of four resblocks followed by a linear layer. All operations related to GMM were performed on the 512-dimensional latent space (i.e., output of the fourth resblock). Number of EM iterations to optimize GMM parameters $\psi$ is set to 10.

- During **supervised learning** stage, a batch size of 100 was used, and training was conducted for 200 epochs. The learning rate was initialized to be 0.01 and decreased to 0.001 at the $160^{th}$ epoch. Cross-entropy loss was adopted for training the classifier, same as the other works. Additionally, an SGD optimizer with momentum of 0.9 and a decay rate of 5e-4 were applied to train the classifier. Aside from a minibatch of size 100 used for the backpropagation, we additionally sample a balanced subset $X_s$ of size 1,000 from the labeled set solely for the computation of GMM statistics $\psi_L$ (e.g., for CIFAR10, $X_s$ contains 100 samples for each of 10 classes; for CIFAR100, $X_s$ contains 10 samples for each of 100 classes). In cases where there are fewer than 10 labeled samples in a minor class (e.g., during early AL cycles in CIFAR-100), we randomly select existing labeled samples and augment them through random cropping/horizontal-flipping to construct a balanced set $X_s$. GMM is fitted to $X_s$ to get $\psi_s$ and this $\psi_s$ replaces $\psi_L$ in $L_{GMM}$. We cut the gradient graph to GMM statistics $\psi$ when computing the regularization loss $L_{GMM}$, preventing gradients from flowing back through it during backpropagation to reduce memory and computational requirements. Additionally, the balanced subset $X_s$ has labels assigned to it, allowing us to directly compute class-mean $\mu$ and class-variance $\sigma^2$ without a demand of computation for EM algorithm. The constant weight $\alpha$ for GMM loss with respect to cross-entropy loss is set to 0.0001.

- During **semi-supervised learning** stage, the task model and the discriminator $D_\phi$ are trained for a total of 10,000 iterations, where we sampled 100 labeled and unlabeled samples in each iteration. We used the Adam optimizer to train the discriminator with an initial learning rate of 5e-4 that is decayed by the factor of 0.1 at the $8,000^{th}$ iteration. Throughout the 10,000 iterations, we computed GMM statistics $\psi_L, \psi_{UL}$ using the entire dataset at every $500^{th}$ iteration. It is noteworthy that although GMM statistics $\psi$ were computed using a subset $X_s$ in every iteration during the supervised learning stage, the statistics $\psi$ were calculated less frequently but using the entire dataset during the semi-supervised learning stage. It is because the neural network becomes much more stable and updates of model parameters become less drastic after the supervised learning stage. This approach significantly reduced computational complexity while still obtaining accurate distributional information by performing GMM fitting using all available data. The constant weight $\alpha$ for discriminator loss (i.e, $L_{Dis} = D_\phi(z_u l) - D_\phi(z_l)$) with respect to cross-entropy loss is 1.

## A.2 Algorithm Description

---

**Algorithm 1** Proposed Method : Distribution Aware Active Learning via Gaussian Mixtures

---

**Input:** Initialized task model $f_\theta$, initialized discriminator model $D_\phi$, initial labeled set $X_L^0$ and unlabeled set $X_{UL}^0$, constant weight $\alpha, \beta$, number of category $K$, iteration $N$ of the semi-supervised learning stage, interval $M$ to fit GMM during the semi-supervised learning stage

**Output:** Trained models $f_\theta^T, D_\phi^T$ after $T$ active learning cycles

**for** each active learning cycle $t = 0, 1, ..., T-1$ **do**

**Step 1: Supervised learning stage to train $f_\theta^t$ using $X_L^t$**

1:   **for** each minibatch $B \subset X_L^t$ **do**
2:      Draw an auxiliary balanced subset $X_s \subset X_L^t$
3:      Optimize GMM parameters $\psi_s := (\pi_s, \mu_s, \sigma_s^2)$ using $X_s^t$
4:      Compute cross-entropy loss $\mathbb{E}_{x_l \in B} \left[ L_{CE}(f_\theta^t, x_l) \right]$
5:      Compute GMM-based NLL loss $\mathbb{E}_{x_l \in B} \left[ L_{GMM}(f_\theta^t, x_l; \psi_s) \right]$
6:      **Train** $f_\theta^t$ to minimize $\mathbb{E}_{x_l \in B} \left[ L_{CE}(f_\theta^t, x_l) + \alpha L_{GMM}(f_\theta^t, x_l; \psi_s) \right]$
7:   **end for**

**Step 2: Semi-supervised learning stage to train $f_\theta^t, D_\phi^t$ using $X_L^t, X_{UL}^t$**

1:   **for** each each iteration $n = 1, ..., N$ **do**
2:      **if** $n \mod M = 0$ **then**
3:         Optimize GMM parameters $\psi_L, \psi_{UL}$ using $X_L^t, X_{UL}^t$
4:      **end if**
5:      Draw batches $B_L, B_{UL}$ respectively from $X_L^t, X_{UL}^t$
6:      Compute cross-entropy loss $\mathbb{E}_{x_l \in B_L} \left[ L_{CE}(f_\theta^t, x_l) \right]$
7:      Compute discriminator loss $\mathbb{E}_{x_l \in B_L, x_{ul} \in B_{UL}} \left[ D_\phi^t(x_{ul}) - D_\phi^t(x_l) \right]$ w.r.t. $f_\theta^t$
8:      **Train** $f_\theta^t$ to minimize $\mathbb{E}_{x_l \in B_L, x_{ul} \in B_{UL}} \left[ L_{CE}(f_\theta^t, x_l) + D_\phi^t(x_{ul}) - D_\phi^t(x_l) \right]$
9:      Compute discriminator loss $\mathbb{E}_{x_l \in B_L, x_{ul} \in B_{UL}} \left[ D_\phi^t(x_l) - D_\phi^t(x_{ul}) \right]$ w.r.t. $D_\phi^t$
10:     **Train** $D_\phi^t$ to minimize $\mathbb{E}_{x_l \in B_L, x_{ul} \in B_{UL}} \left[ D_\phi^t(x_l) - D_\phi^t(x_{ul}) \right]$
11: **end for**

**Step 3: Data selection stage to choose informative data and update $X_L^t, X_{UL}^t$**

1:   Compute informativeness $I_{Like}(x_{ul}), I_{Dis}(x_{ul}), I_{Ent}(x_{ul})$ for $x_{ul} \in X_{UL}^t$
2:   Apply rank-normalized informativeness
    $I_{total} = Rank(I_{Like}(x_{ul})) + Rank(I_{Dis}(x_{ul})) + Rank(I_{Ent}(x_{ul}))$
3:   Select the most informative data $I^t \subset X_{UL}^t$ based on $I_{total}$
4:   Human oracles annotates $I^t$
5:   Update labeled set, $X_L^{t+1} = X_L^t \cup I^t$
6:   Update unlabeled set, $X_{UL}^{t+1} = X_{UL}^t n I^t$

---

The detailed procedure of the proposed **Distribution-Aware Active Learning via Gaussian Mixtures** is given in Algorithm 1. Proposed algorithm consists of two learning stages (supervised, semi-supervised) and one selection stage. Both of these learning stages commonly utilize GMM statistics $\psi$. During the supervised learning stage, in addition to calculating the CE loss using the data batch $B$, a regularization loss is computed using $\psi_s$ obtained from a balanced subset $X_s$. This regularization loss helps improve the robustness of the task model $f_\theta$ against noisy local batches by incorporating supervision based on the global data distribution $\psi$. Semi-supervised learning reduces the distance between the latent spaces of $X_L$ and $X_{UL}$ through adversarial training between $f_\theta$ and $D_\phi$. Our proposed method first computes GMM likelihood and GMM posteriors using $\psi_L, \psi_{UL}$ and feeds them into $D_\phi$, preventing $D_\phi$ from memorizing individual samples and enabling consideration of general distributional information.

## A.3 Ablation Studies on Informativeness Metrics

In the proposed selection strategy, we measure the informativeness of unlabeled samples by $I_{total}$ that is the summation of three rank-normalized informativeness metrics ($I_{Like}, I_{Dis}, I_{Ent}$). In this section, to gain insights into the combination of three informativeness metrics, we conduct ablation studies by varying the combinations and report the corresponding results in Fig. 13. All experimental settings are same as in the main paper, with the only difference being the resulting informativeness metric $I_{total}$. At most cycles, combinations of metrics tend to show better performance than if only single information source is considered. Also, while distribution-aware metric ($I_{Like}, I_{Dis}$) contribute additional improvement than $I_{Ent}$ at early cycles, this trend is reversed at later cycles as the

distributions of $X_L, X_{UL}$ become similar. This result demonstrates that dispositions of each metric are disparate across the learning cycles, confirming the importance of multiple sources to avoid unnecessary redundancy.

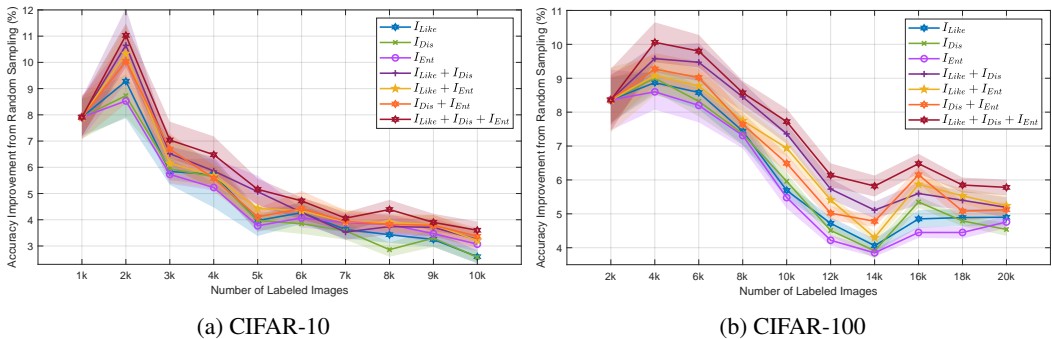

|  |  |
|---|---|
| (a) CIFAR-10 | (b) CIFAR-100 |

Figure 13: Accuracy gain from random sampling by the combination of three informativeness metrics.

## A.4  ADDITIONAL EXPERIEMNTS IN CATEGORY MISMATCH SETTING (UNSEEN CLASSES)

Due to the nature of active learning, small labeled sets may not include all categories of the entire dataset during early learning cycles. In such cases, the task model (i.e., classifier) is trained using only partial category information, and previously unseen classes are gradually added to the labeled set through subsequent learning cycles. In this scenario, the selection strategy of the active learning algorithm plays a crucial role in assigning high informativeness to samples from unseen classes, enabling the rapid incorporation of unseen classes into the labeled set. To verify whether our proposed algorithm exhibits robust performance in realistic settings as described above, we defined a new setting called 'category mismatch.' In this setting, we ensured that only 70% of the total categories were included in the labeled set during the first cycle. Specifically, we included only the labels corresponding to the first 70% of the categories. For example, in the case of CIFAR10, categories 0, 1, 2, ..., 6 would be included in the initial labeled set. We conducted experiments on the CIFAR10 and CIFAR100 datasets to measure the performance of active learning with the category mismatch scenario. in Fig. 14, we report the accuracy improvement from random sampling. It can be observed that the proposed methods outperform the baselines and demonstrate consistent results with those mentioned in the manuscript.

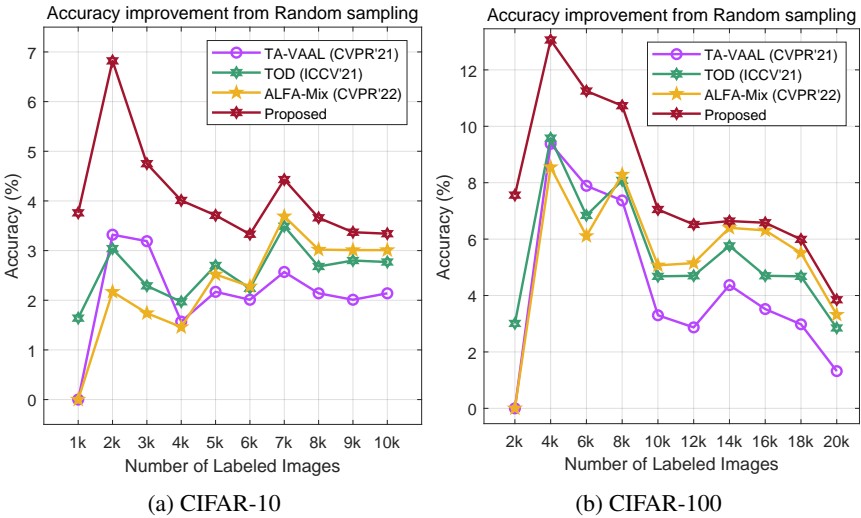

|  |  |
|---|---|
| (a) CIFAR-10 | (b) CIFAR-100 |

Figure 14: Accuracy gain from random sampling by the combination of three informativeness metrics.

The superior performance of the proposed method compared to the baseline can be attributed to the ability to estimate the distribution of the entire dataset, even with a shortage of certain categories in the labeled set, using Gaussian Mixture Models (GMMs) and the intact unlabeled set. The estimated GMM statistics can be effectively used to align the distributions of $X_L$ and $X_{UL}$ when training the model or selecting data. For example, in the first cycle of CIFAR10, the GMM $\psi_L$ of the labeled set may have only seven modes, resulting in a low likelihood for the latent space where three unseen classes exist. On the other hand, the GMM $\psi_{UL}$ of the unlabeled set may have ten modes and a high likelihood for the latent space where unseen classes exist. Therefore, $I_{like}$ predicts high informativeness for unseen classes and helps the distribution of the labeled set to quickly catch up with the distribution of the unlabeled set.

## A.5 ADDITIONAL EXPERIEMNTS IN LONG-TAILED IMBALANCE SETTING

To demonstrate the robustness of our proposed algorithm in various realistic scenarios, we conducted additional experiments in the long-tailed imbalance setting. Following the experimental setup of (Cui et al., 2019; Bengar et al., 2022), we performed experiments on the long-tailed versions of CIFAR10 and CIFAR100 datasets. Specifically, the number of training samples for each category decreases exponentially according to the function $n = n_i \mu^i$, where $n_i$ is the original number of samples for category $i$ and $\mu$ is the imbalance factor, which we chose to be 0.1. We compared the results of our proposed algorithm with CBAL in Fig. 15. CBAL is an active learning method that proposes a class-balancing loss to alleviate imbalance in long-tail distributions. The experimental results show that the proposed methods significantly outperform CBAL, which is consistent with the findings in Fig. 10 of the main text, where the baseline was outperformed in the imbalance setting.

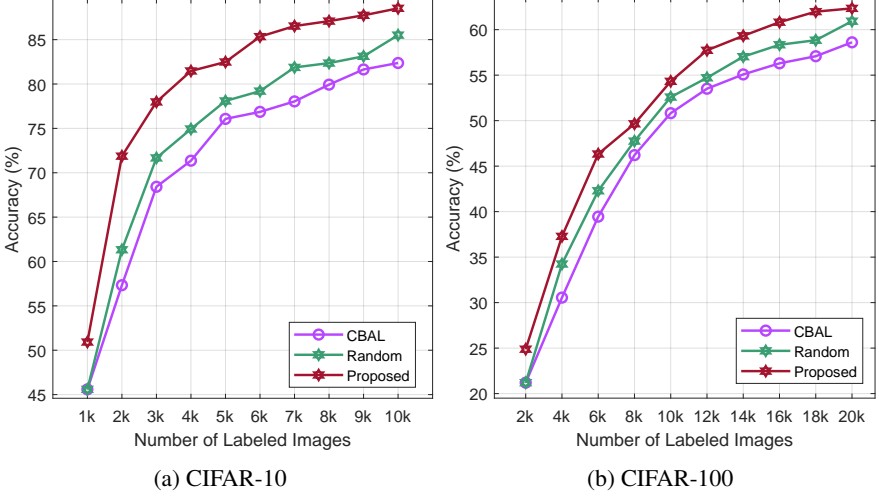

(a) CIFAR-10        (b) CIFAR-100

Figure 15: Performance comparison on long-tailed imbalanced datasets (imbalance factor = 0.1).

Our performance gains are the result of the collaborative action of three proposed schemes. In particular, the distribution-aware regularization estimates the data distribution of the entire labeled set using a Gaussian Mixture Model (GMM) and generates supervisory signals using a distance-based loss calculated from GMM statistics. This helps mitigate the overfitting of the task model to major classes and improves performance. Additionally, the likelihood-based metric $I_{like}$ in equation 4 assigns high informativeness to minor classes, increasing their chances of being selected during the selection process.

## A.6 COMPARISON AND COMPATIBILITY ANALYSIS WITH OTHER DISTRIBUTIONAL ALIGNMENT METHODS

To validate the superiority of the proposed algorithm, we conducted additional comparison experiments with ADS (Fu et al., 2021), which performs distributional alignment through adversarial training, on CIFAR-10 and CIFAR-100 datasets. The results of these experiments are reported in

Fig. 16. Both methods perform distributional alignment through adversarial training, but the larger performance gain in our works is attributed to the fact that our key idea supports distributional alignment using distributional information obtained through GMM, rather than relying solely on adversarial learning for distributional alignment. As shown in Fig.5 5, as can be seen, existing adversarial training tends to overfit the discriminator to small $X_L$, causing it to memorize the labeled/unlabeled samples themselves and consequently generating overconfident predictions (i.e., low entropy), which undermines the reliability of informativeness. On the other hand, the proposed method feeds GMM statistics to the discriminator, allowing it to consider the distributions of $X_L$ and $X_{UL}$ as a whole to predict informativeness. As a result, we observed that the predictions of the discriminator were calibrated, leading to improved performance.

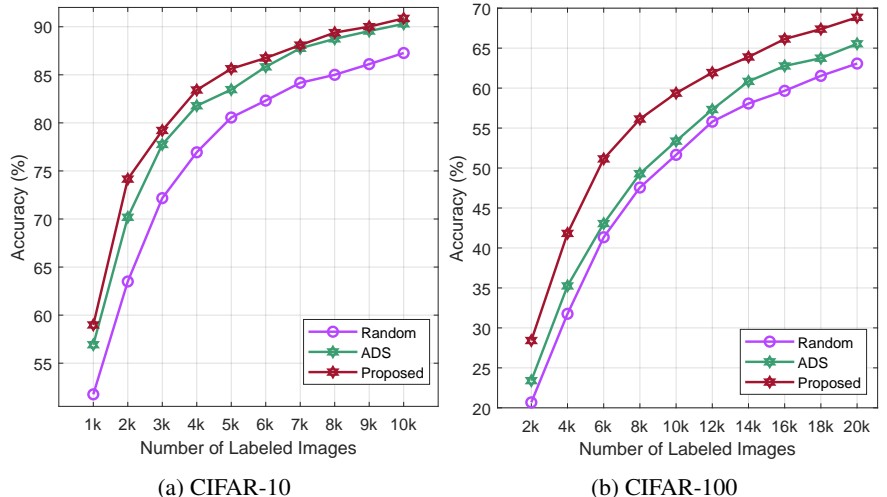

(a) CIFAR-10          (b) CIFAR-100

Figure 16: Performance comparison with ADS (Fu et al., 2021).

To validate that our proposed GMM-aided distributional alignment brings about general performance improvement, we infused GMM information into TA-VAAL and WAAL, which perform alignment through the discriminator in a similar manner. Specifically, we applied the EM algorithm to the output of each method's feature extractor to calculate GMM statistics. We then fed the GMM-based posterior probabilities $\{p(y = k|z; \psi_L)\}_{k=1}^{K}$, $\{p(y = k|z; \psi UL)\}_{k=1}^{K}$, and likelihoods $p(z; \psi_L)$, $p(z; \psi UL)$ to the discriminator using the approach shown in Fig. 4. In Fig. 17, it is observed that the discriminator outputs higher prediction entropy when using GMM-based distributional information. This is because the discriminator considers the overall distributional information rather than memorizing labeled samples themselves when making decisions. As a result, we can confirm that both accuracy and performance are further improved.

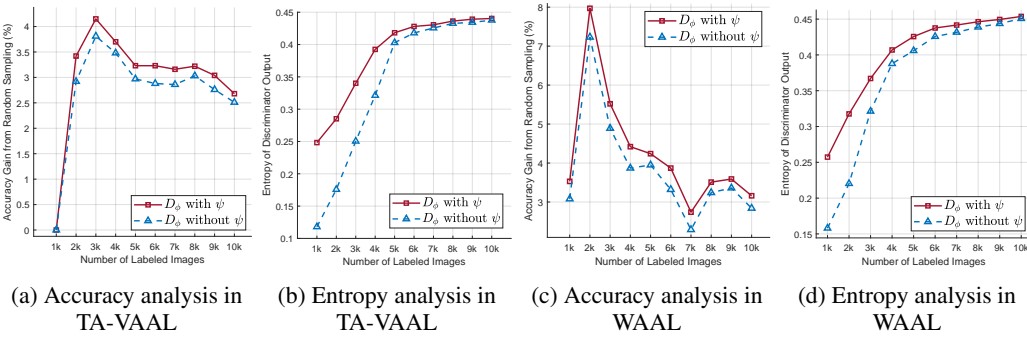

(a) Accuracy analysis in TA-VAAL    (b) Entropy analysis in TA-VAAL    (c) Accuracy analysis in WAAL    (d) Entropy analysis in WAAL

Figure 17: compatibility analyis of the proposed infusion of GMM statistics with adversarial active learning works (TA-VAAL, WAAL) on CIFAR-10

A.7 ADDITIOINAL EXPERIMENTS USING RESNET50 AND TINYIMAGENET

To further validate the generality of the proposed algorithm and GMM, we conducted additional experiments in more challenging settings. We reported the experimental results using ResNet50 in Fig. 18, and the experimental results using ResNet18 backbone on TinyImageNet in Fig. 19. We used a batch size of 100 and trained for 200 epochs, following the experimental settings of (Jung et al., 2022), using the OneCycleLR scheduler with a max_lr of 0.1 and a div_factor of 25. To reduce computational burden, we performed forward propagation, including the computation of GMM statistics, using PyTorch's automatic mixed precision, with floating point 16. The experimental results were consistent with the results in the main paper, confirming the general superiority of the proposed algorithm.

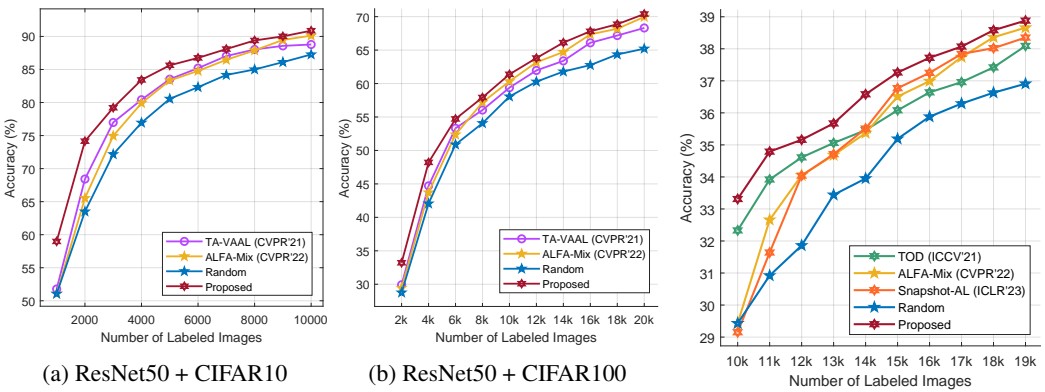

(a) ResNet50 + CIFAR10     (b) ResNet50 + CIFAR100

Figure 18: Comparison with other baslines using ResNet50.     Figure 19: ResNet18 + CIFAR100

A.8 TIME COMPLEXITY ABOUT THE COMPUTATION OF GMM

In this work, GMM $\psi$ is computed in both the "supervised learning stage" and the "semi-supervised learning stage" and GMM calculations impose an increased computational burdens of approximately 5.66% and 15.19% in each respective stage. For detailed explanations, refer to the information provided below.

**Supervised learning stage.** We calculate the GMM parameters $\psi_L$ for the labeled set ($X_L$) once per minibatch (we do not compute the GMM for $X_{UL}$ at this stage). As we have label information, we can directly compute $\psi_s = (\pi_s, \mu_s, \sigma_s^2)$ without multiple iterations of the EM algorithm. Detailed analysis on the operational time can be seen in Table 2.

| Remarks | Method | Time required |
|---|---|---|
| — | Draw a minibatch ($B$) from the dataloader | 33.1505 msec. |
| GMM computation | Draw a i.i.d. subset ($X_s$) from the dataloader | 4.5133 msec. |
| | Pass $X_s$ through the feature extractor $f_\theta$ to get $f_\theta(X_s)$ | 6.1559 msec. |
| | Caclulate GMM $\psi_s$ from $f_\theta(X_s)$ | 0.0470 msec. |
| — | Compute $Loss_{CE}, Loss_{GMM}$ using minibatch ($B$) and GMM $\psi_s$ | 3.7744 msec. |
| — | Update model by minimizing $Loss_{CE}, Loss_{GMM}$ | 141.7344 msec. |

Table 2: Analysis of the required time to process one batch in the supervised learning stage.

**Semi-supervised learning stage.**: This stage consists of a total of 10,000 iterations (batches), and GMMs $\psi_L, \psi_{UL}$ are computed at the every 500 iterations for semi-supervised learning. Similarly with the above, $\psi_L$ can be directly computed using the labels and only $\psi_{UL}$ is computed via an iterative EM algorithm. The computation of GMM accounts for 15.19% of the entire time required for semi-supervised learning, and a detailed time analysis is provided in Table 3

| Method | Time required |
|---|---|
| Time required for 500 iterations (including GMM calculation) | 144.9939 sec. |
| Time required for GMM calculation | 22.0256 sec. |
| Ratio of time required for GMM calculation during semi-supervised learning | 15.1906% |

Table 3: Analysis of the required time to process one batch in the semi-supervised learning stage.

**Thoretical analysis about time complexity of GMM computation.** Basically, the time complexity for the computation of vanilla GMM is $O(NKID^2)$, where $N, K, I, D$ represent the number of data, number of Gaussian modes, number of EM iterations, and the dimensionality of the data. However, we reduced the complexity to $O(NKID)$ by making the isotropic Gaussian assumption, which helps accelerate GMM computation. Moreover, as evident from the above experimental analysis, the time required for GMM computation is not significantly demanding compared to the time spent on forward/backward propagation for model update.

### A.9 ANALYSIS OF HYPERPARAMETERS THAT BALANCE MULTIPLE LOSSES

The value of hyperparameter $\alpha$ that balances $Loss_{CE}$ and $Loss_{GMM}$ is fixed to be 1e-4, and details on hyperparameter including $\alpha$ can be seen in Section A.2. We compared the performance by varying the value of $\alpha$ from 0.00001 to 0.001, and reported the results below. When α is too large, distributional regularization on the latent space compromises the performance of the main task. On the other hand, when α is too small, the effect of regularization and performance improvement diminishes.

| Value of $\alpha$ | Accuracy (%) on Proportion (%) of Labeled Samples | | | | | | | | | |
|---|---|---|---|---|---|---|---|---|---|---|
| | 2.0 | 4.0 | 6.0 | 8.0 | 10.0 | 12.0 | 14.0 | 16.0 | 18.0 | 20.0 |
| 0.001 | 59.24 | 73.96 | 77.81 | 82.81 | 85.22 | 85.88 | 87.60 | 88.13 | 88.96 | 89.53 |
| 0.0005 | 59.21 | 74.07 | 78.98 | 82.73 | 85.40 | 86.67 | 87.57 | 89.04 | 89.57 | 90.19 |
| 0.0001 | 60.64 | 74.15 | 79.21 | 83.42 | 85.63 | 86.75 | 88.09 | 89.39 | 90.01 | 90.87 |
| 0.00005 | 58.98 | 73.63 | 77.45 | 82.16 | 84.76 | 85.88 | 87.43 | 88.95 | 89.63 | 90.38 |
| 0.00001 | 59.58 | 73.41 | 76.52 | 81.78 | 84.25 | 86.22 | 87.32 | 88.83 | 89.44 | 90.01 |

Table 4: Analysis of the effect of $\alpha$ to the performance on CIFAR10

As for the loss functions during adversarial training, we use two losses : cross-entropy loss $L_{CE}$ and discriminator loss $D_\phi(x_{ul}) - D_\phi(x_l)$. Without a special constant weight to balance them, we added two losses in a 1:1 ratio and conducted the adversarial semi-supervised leraning.

### A.10 T-SNE VISUALIZATION OF SELECTIONS FROM VARIOUS AL METHODS

In Fig.20, we illustrate the intuitive patterns of candidate set selected by various AL methods via t-SNE visualization. Before t-SNE, we first train a feature extractor and classifier using randomly selected 2,000 samples from imbalanced CIFAR10 (imbalance ratio = 10). Afterward, we depict the candidate set selected by various AL methods. From the observations, we can take a hint of properties that various AL method have for selection. For instance, Snapshot (Jung et al., 2022) that calculates uncertainty using the output of classifier exhibits redundancy by selecting multiple samples from narrow regions. Coreset that considers only diversity can select diverse samples in broad areas but it lacks the ability to specify certain informative regions. CoreGCN that prefers samples expected be from unlabeled set, is observed to select samples from already familiar areas with dense labeled samples, as well as outlier samples that hardly represent $X_{UL}$. In contrast, the candidate set from the proposed method demonstrates desired properties: 1) it is diverse enough to encompass broad area while 2) represents the distribution of $X_{UL}$ well. Our method compares GMMs of $X_L, X_{UL}$ to select samples that are unfamiliar in $X_L$, but representative of $X_{UL}$; especially in the imbalanced setting, this comparison leads to select minor classes that are rare in $X_L$, thereby contributing to the significant performance gain as seen in Fig.10

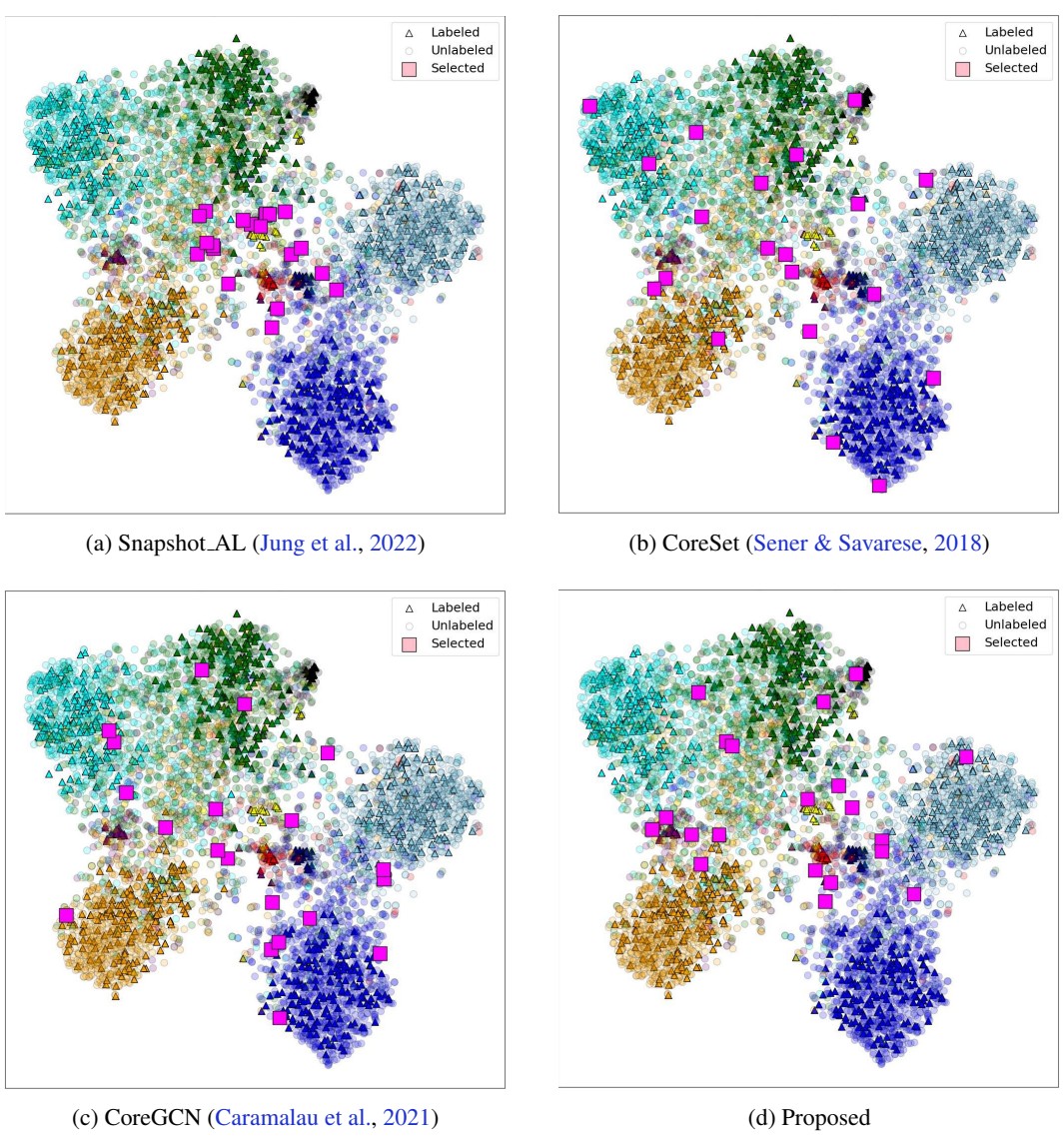

(a) Snapshot_AL (Jung et al., 2022)

(b) CoreSet (Sener & Savarese, 2018)

(c) CoreGCN (Caramalau et al., 2021)

(d) Proposed

Figure 20: t-SNE visualization of selected samples by various AL methods. t-SNE was applied to the latent space from imbalanced CIFAR10 (imbalance ratio = 10). Colors of markers represent the category of samples, and markers $\triangle, \bigcirc, \square$ denote labeled samples, unlabeled samples and selected samples, respectively. For every AL method, 20 selected samples $\square$ are depicted.

## A.11 PRECISE ACCURACIES WITH MORE BASELINES

In the main paper, we displayed the performance of only a subset of baselines (WAAL (Shui et al., 2020), TA-VAAL (Kim et al., 2021), CoreGCN (Caramalau et al., 2021), TOD (Huang et al., 2021), ALFA-MIX (Parvaneh et al., 2022), SnapshotAL (Jung et al., 2022)) in line graph format for readability. In this section, we additionally report the performance of Coreset (Sener & Savarese, 2018), VAAL (Kim et al., 2021), and LL4AL (Yoo & Kweon, 2019) in tabular form, which was not previously reported in the main paper. The experimental settings, including task model architecture, data augmentation methods, hyperparameter settings, and annotation budget per learning cycle, are consistent with those described in the main paper.

| Methods | Accuracy (%) on Proportion (%) of Labeled Samples | | | | | | | | | | |
|---|---|---|---|---|---|---|---|---|---|---|---|
| | 2.0 | 4.0 | 6.0 | 8.0 | 10.0 | 12.0 | 14.0 | 16.0 | 18.0 | 20.0 | Avg |
| Random | 51.76 | 63.50 | 72.17 | 76.94 | 80.56 | 82.32 | 84.17 | 85.00 | 86.11 | 87.27 | 76.98 |
| CoreSet | 51.76 | 66.56 | 75.94 | 80.44 | 82.39 | 85.00 | 86.61 | 88.06 | 88.89 | 89.67 | 79.53 |
| LL4AL | 51.06 | 62.86 | 74.49 | 79.21 | 81.69 | 83.57 | 86.71 | 88.21 | 88.56 | 89.24 | 78.56 |
| VAAL | 51.06 | 64.28 | 72.50 | 77.94 | 81.17 | 82.83 | 84.93 | 85.71 | 87.15 | 87.63 | 77.52 |
| Snapshot-AL | 51.76 | 63.42 | 73.44 | 79.31 | 82.10 | 84.57 | 85.91 | 87.34 | 88.15 | 88.98 | 78.49 |
| TA-VAAL | 51.78 | 66.42 | 75.98 | 80.42 | 83.53 | 85.20 | 87.03 | 88.03 | 88.87 | 89.78 | 79.70 |
| CoreGCN | 51.76 | 62.35 | 73.04 | 79.80 | 82.79 | 84.49 | 86.88 | 88.07 | 88.71 | 89.66 | 78.76 |
| TOD | 54.19 | 68.37 | 76.42 | 80.28 | 83.57 | 85.76 | 87.00 | 88.57 | 89.00 | 90.02 | 80.31 |
| ALFA-Mix | 51.76 | 65.53 | 74.95 | 79.91 | 83.34 | 84.78 | 86.46 | 87.89 | 89.47 | 90.12 | 79.42 |
| WAAL | 54.86 | 70.73 | 77.06 | 80.81 | 84.51 | 85.64 | 86.48 | 88.24 | 89.47 | 90.11 | 80.79 |
| **Proposed** | **58.98** | **74.15** | **79.21** | **83.42** | **85.63** | **86.75** | **88.09** | **89.39** | **90.01** | **90.87** | **82.65** |

Table 5: Results on balanced CIFAR-10.

| Methods | Accuracy (%) on Proportion (%) of Labeled Samples | | | | | | | | | | |
|---|---|---|---|---|---|---|---|---|---|---|---|
| | 2.0 | 4.0 | 6.0 | 8.0 | 10.0 | 12.0 | 14.0 | 16.0 | 18.0 | 20.0 | Avg |
| Random | 20.68 | 31.76 | 41.34 | 47.56 | 51.65 | 55.80 | 58.07 | 59.66 | 61.53 | 63.07 | 49.11 |
| CoreSet | 20.68 | 31.16 | 42.67 | 49.03 | 54.94 | 58.35 | 61.08 | 63.30 | 64.89 | 66.08 | 51.22 |
| LL4AL | 20.68 | 30.80 | 40.85 | 49.26 | 54.83 | 58.98 | 61.65 | 63.13 | 65.23 | 66.53 | 51.19 |
| VAAL | 20.68 | 33.13 | 42.27 | 50.23 | 54.60 | 57.24 | 59.97 | 62.01 | 63.72 | 65.36 | 50.92 |
| Snapshot-AL | 20.68 | 34.13 | 45.26 | 51.32 | 56.46 | 59.02 | 62.71 | 64.18 | 65.56 | 67.97 | 52.73 |
| TA-VAAL | 20.68 | 31.99 | 42.05 | 50.68 | 56.08 | 59.49 | 62.33 | 63.92 | 66.32 | 67.74 | 52.13 |
| CoreGCN | 20.86 | 36.54 | 44.85 | 52.61 | 57.22 | 60.41 | 62.57 | 64.54 | 66.41 | 67.06 | 53.31 |
| TOD | 23.42 | 35.91 | 45.37 | 51.91 | 56.03 | 58.74 | 61.43 | 63.80 | 65.22 | 67.14 | 52.90 |
| ALFA-Mix | 20.68 | 33.02 | 44.14 | 52.00 | 57.38 | 59.91 | 63.29 | 65.47 | 66.66 | 68.68 | 53.12 |
| WAAL | 24.24 | 39.10 | 47.05 | 52.02 | 56.89 | 58.36 | 61.78 | 63.86 | 64.70 | 65.70 | 53.37 |
| **Proposed** | **28.42** | **41.82** | **51.14** | **56.13** | **59.37** | **61.94** | **63.89** | **66.14** | **67.38** | **68.85** | **56.51** |

Table 6: Results on balanced CIFAR-100.

| Methods | Accuracy (%) on Proportion (%) of Labeled Samples | | | | | | | | | | |
|---|---|---|---|---|---|---|---|---|---|---|---|
| | 2.0 | 4.0 | 6.0 | 8.0 | 10.0 | 12.0 | 14.0 | 16.0 | 18.0 | 20.0 | Avg |
| Random | 82.56 | 87.89 | 89.73 | 91.04 | 91.86 | 92.40 | 92.67 | 93.22 | 93.50 | 94.00 | 90.89 |
| CoreSet | 82.56 | 88.43 | 90.48 | 91.72 | 92.79 | 93.36 | 93.75 | 94.08 | 94.32 | 94.79 | 91.63 |
| LL4AL | 82.56 | 88.04 | 89.50 | 91.24 | 92.26 | 92.81 | 93.51 | 94.13 | 94.12 | 94.73 | 91.29 |
| VAAL | 82.56 | 88.68 | 90.69 | 91.82 | 92.71 | 93.30 | 93.78 | 94.18 | 94.48 | 94.89 | 91.71 |
| Snapshot-AL | 82.56 | 89.04 | 90.86 | 92.13 | 92.95 | 93.61 | 94.31 | 94.57 | 94.96 | 95.36 | 92.04 |
| TA-VAAL | 82.56 | 88.39 | 90.39 | 91.61 | 92.98 | 93.66 | 93.99 | 94.13 | 94.36 | 94.85 | 91.69 |
| CoreGCN | 82.56 | 89.13 | 91.02 | 91.92 | 92.75 | 93.48 | 94.09 | 94.44 | 94.84 | 95.17 | 91.94 |
| TOD | 83.43 | 88.83 | 91.63 | 93.04 | 93.70 | 94.09 | 94.70 | 94.96 | 95.19 | 95.46 | 92.50 |
| ALFA-Mix | 82.56 | 89.01 | 91.31 | 92.51 | 93.52 | 94.02 | 94.58 | 94.88 | 95.10 | 95.51 | 92.30 |
| WAAL | 84.28 | 89.42 | 91.80 | 92.81 | 93.62 | 94.18 | 94.76 | 95.01 | 95.15 | 95.34 | 92.64 |
| **Proposed** | **85.62** | **90.19** | **92.00** | **93.57** | **94.40** | **94.66** | **95.04** | **95.31** | **95.67** | **95.86** | **93.23** |

Table 7: Results on balanced SVHN.

| Methods | Accuracy (%) on Proportion (%) of Labeled Samples | | | | | | | | | | |
|---|---|---|---|---|---|---|---|---|---|---|---|
| | 2.0 | 4.0 | 6.0 | 8.0 | 10.0 | 12.0 | 14.0 | 16.0 | 18.0 | 20.0 | Avg |
| Random | 81.73 | 87.11 | 89.07 | 89.69 | 89.92 | 90.26 | 90.56 | 90.94 | 91.28 | 91.53 | 89.21 |
| CoreSet | 81.73 | 86.97 | 89.59 | 90.24 | 90.40 | 90.94 | 91.24 | 91.95 | 92.49 | 92.45 | 89.80 |
| LL4AL | 81.73 | 85.90 | 88.40 | 89.40 | 90.10 | 90.56 | 91.02 | 91.49 | 91.85 | 92.06 | 89.25 |
| VAAL | 81.73 | 87.40 | 89.54 | 90.28 | 90.73 | 91.22 | 91.55 | 92.01 | 92.37 | 92.52 | 89.93 |
| Snapshot-AL | 81.73 | 86.57 | 88.75 | 90.25 | 91.13 | 91.61 | 92.32 | 92.78 | 93.01 | 93.11 | 90.13 |
| TA-VAAL | 81.73 | 86.61 | 89.74 | 90.25 | 91.31 | 91.72 | 91.93 | 92.34 | 92.80 | 92.86 | 90.13 |
| CoreGCN | 81.73 | 86.30 | 88.78 | 90.14 | 90.95 | 91.39 | 91.78 | 92.27 | 92.44 | 92.74 | 89.85 |
| TOD | 83.52 | 88.51 | 89.57 | 91.17 | 91.81 | 92.55 | 92.77 | 93.10 | 93.16 | 93.37 | 90.95 |
| ALFA-Mix | 81.73 | 88.13 | 89.95 | 90.91 | 91.88 | 92.46 | 92.84 | 93.13 | 93.33 | 93.57 | 90.79 |
| WAAL | 84.20 | 88.87 | 90.14 | 91.34 | 91.51 | 92.27 | 92.46 | 93.08 | 93.11 | 93.37 | 91.04 |
| **Proposed** | **85.31** | **89.51** | **90.67** | **91.52** | **92.29** | **92.79** | **93.03** | **93.19** | **93.48** | **93.69** | **91.55** |

Table 8: Results on balanced FashionMNIST.

| Methods | Accuracy (%) on Proportion (%) of Labeled Samples | | | | | | | | | | |
|---|---|---|---|---|---|---|---|---|---|---|---|
| | 2.0 | 4.0 | 6.0 | 8.0 | 10.0 | 12.0 | 14.0 | 16.0 | 18.0 | 20.0 | Avg |
| Random | 35.50 | 47.98 | 54.83 | 62.57 | 66.27 | 69.90 | 72.66 | 74.15 | 75.92 | 77.11 | 63.69 |
| CoreSet | 35.50 | 50.33 | 58.88 | 66.27 | 70.18 | 73.85 | 77.08 | 79.46 | 81.09 | 81.81 | 67.45 |
| LL4AL | 35.50 | 51.94 | 58.59 | 65.81 | 70.21 | 74.85 | 78.28 | 80.01 | 80.84 | 82.06 | 67.81 |
| VAAL | 35.50 | 51.64 | 58.97 | 66.32 | 70.87 | 74.50 | 77.35 | 79.53 | 81.01 | 81.65 | 67.74 |
| Snapshot-AL | 35.50 | 53.18 | 61.04 | 67.16 | 71.88 | 75.91 | 78.34 | 82.01 | 83.46 | 84.06 | 69.25 |
| TA-VAAL | 35.50 | 50.68 | 59.02 | 65.69 | 71.23 | 75.98 | 78.89 | 81.73 | 82.98 | 83.49 | 68.52 |
| CoreGCN | 35.50 | 49.83 | 60.77 | 69.08 | 72.40 | 75.75 | 80.25 | 82.23 | 83.90 | 84.27 | 69.40 |
| TOD | 42.02 | 57.75 | 65.59 | 70.01 | 71.42 | 74.62 | 76.48 | 79.42 | 80.74 | 81.84 | 69.99 |
| ALFA-Mix | 35.50 | 56.25 | 63.06 | 70.71 | 75.11 | 77.63 | 80.51 | 82.72 | 84.13 | 84.35 | 71.00 |
| WAAL | 43.08 | 59.57 | 68.59 | 72.18 | 75.12 | 78.26 | 79.60 | 80.83 | 81.22 | 81.45 | 71.99 |
| **Proposed** | **46.37** | **66.48** | **74.57** | **78.33** | **80.61** | **82.67** | **83.64** | **84.78** | **84.95** | **85.38** | **76.78** |

Table 9: Results on imbalanced CIFAR10.

| Methods | Accuracy (%) on Proportion (%) of Labeled Samples | | | | | | | | | | |
|---|---|---|---|---|---|---|---|---|---|---|---|
| | 2.0 | 4.0 | 6.0 | 8.0 | 10.0 | 12.0 | 14.0 | 16.0 | 18.0 | 20.0 | Avg |
| Random | 11.31 | 16.20 | 21.61 | 25.44 | 29.18 | 32.83 | 35.18 | 37.10 | 39.17 | 39.76 | 28.78 |
| CoreSet | 11.31 | 17.01 | 23.30 | 27.48 | 31.76 | 34.96 | 37.85 | 40.32 | 42.88 | 43.73 | 31.06 |
| LL4AL | 11.31 | 17.71 | 21.73 | 27.05 | 30.54 | 35.58 | 37.96 | 40.42 | 42.80 | 44.27 | 30.94 |
| VAAL | 11.31 | 17.17 | 22.39 | 26.79 | 30.58 | 35.44 | 37.77 | 40.45 | 42.50 | 43.44 | 30.79 |
| Snapshot-AL | 11.31 | 17.05 | 22.65 | 26.77 | 31.01 | 35.51 | 38.42 | 41.45 | 43.95 | 45.21 | 31.33 |
| TA-VAAL | 11.31 | 17.41 | 22.70 | 28.05 | 32.28 | 35.37 | 38.68 | 41.40 | 44.81 | 45.55 | 31.76 |
| CoreGCN | 11.31 | 17.41 | 25.60 | 28.94 | 33.83 | 36.69 | 39.70 | 42.45 | 44.67 | 45.88 | 32.65 |
| TOD | 12.66 | 19.37 | 25.20 | 30.56 | 34.21 | 37.03 | 39.96 | 42.26 | 44.52 | 45.72 | 33.15 |
| ALFA-Mix | 11.31 | 17.61 | 23.83 | 27.89 | 32.01 | 37.91 | 40.18 | 43.84 | 45.54 | 46.30 | 32.64 |
| WAAL | 13.25 | 21.05 | 26.18 | 31.56 | 34.61 | 37.89 | 40.87 | 43.15 | 44.49 | 45.53 | 33.86 |
| **Proposed** | **15.11** | **23.51** | **27.60** | **32.70** | **35.86** | **39.39** | **41.46** | **44.14** | **45.84** | **46.10** | **35.17** |

Table 10: Results on imbalanced CIFAR100.

| Methods | Accuracy (%) on Proportion (%) of Labeled Samples | | | | | | | | | | |
|---|---|---|---|---|---|---|---|---|---|---|---|
| | 2.0 | 4.0 | 6.0 | 8.0 | 10.0 | 12.0 | 14.0 | 16.0 | 18.0 | 20.0 | Avg |
| Random | 62.84 | 69.51 | 72.33 | 76.91 | 79.48 | 81.32 | 81.61 | 83.83 | 84.20 | 84.65 | 77.67 |
| CoreSet | 62.84 | 76.07 | 80.16 | 83.27 | 85.47 | 86.50 | 87.20 | 88.30 | 88.62 | 89.02 | 82.75 |
| LL4AL | 62.84 | 75.84 | 83.28 | 84.99 | 86.86 | 86.87 | 87.73 | 87.93 | 87.96 | 88.72 | 83.30 |
| VAAL | 62.84 | 76.30 | 81.12 | 83.52 | 85.45 | 86.50 | 87.03 | 87.92 | 88.12 | 88.64 | 82.75 |
| Snapshot-AL | 62.84 | 79.49 | 83.43 | 86.13 | 88.43 | 89.31 | 90.67 | 91.34 | 91.54 | 92.13 | 85.53 |
| TA-VAAL | 62.84 | 77.71 | 82.70 | 85.79 | 87.58 | 88.99 | 89.49 | 90.36 | 90.53 | 90.74 | 84.67 |
| CoreGCN | 62.84 | 79.20 | 84.72 | 86.78 | 88.51 | 88.87 | 89.33 | 89.74 | 90.13 | 90.29 | 85.04 |
| TOD | 62.83 | 81.51 | 84.27 | 86.02 | 87.04 | 87.25 | 88.46 | 88.91 | 89.95 | 90.07 | 84.63 |
| ALFA-Mix | 62.84 | 78.60 | 86.57 | 87.82 | 89.75 | 91.17 | 91.57 | 91.83 | 92.36 | 92.64 | 86.52 |
| WAAL | 65.34 | 83.56 | 87.76 | 88.66 | 90.02 | 91.32 | 91.76 | 92.01 | 92.21 | 92.56 | 87.52 |
| **Proposed** | **69.79** | **86.08** | **90.32** | **91.72** | **91.88** | **92.40** | **92.53** | **92.75** | **92.94** | **93.31** | **89.37** |

Table 11: Results on imbalanced SVHN.

| Methods | Accuracy (%) on Proportion (%) of Labeled Samples | | | | | | | | | | |
|---|---|---|---|---|---|---|---|---|---|---|---|
| | 2.0 | 4.0 | 6.0 | 8.0 | 10.0 | 12.0 | 14.0 | 16.0 | 18.0 | 20.0 | Avg |
| Random | 72.24 | 73.48 | 77.39 | 80.16 | 81.62 | 82.91 | 83.30 | 83.93 | 84.19 | 84.27 | 80.35 |
| CoreSet | 72.24 | 79.88 | 83.86 | 85.62 | 86.50 | 87.52 | 87.84 | 88.11 | 88.48 | 88.61 | 84.87 |
| LL4AL | 72.24 | 77.30 | 81.76 | 83.80 | 85.42 | 86.21 | 87.00 | 87.31 | 87.81 | 88.50 | 83.73 |
| VAAL | 72.24 | 76.92 | 81.10 | 83.48 | 84.80 | 85.95 | 86.40 | 86.88 | 87.21 | 87.55 | 83.25 |
| Snapshot-AL | 72.24 | 82.61 | 86.61 | 87.53 | 88.13 | 89.43 | 89.66 | 89.77 | 90.43 | 90.66 | 86.71 |
| TA-VAAL | 72.24 | 79.24 | 83.47 | 85.63 | 86.62 | 87.53 | 88.15 | 88.40 | 88.50 | 89.18 | 84.90 |
| CoreGCN | 72.24 | 79.99 | 84.16 | 86.49 | 87.37 | 88.72 | 88.90 | 89.39 | 89.63 | 89.88 | 85.68 |
| TOD | 73.55 | 83.35 | 86.65 | 87.45 | 87.99 | 88.32 | 88.43 | 88.74 | 89.00 | 89.08 | 86.26 |
| ALFA-Mix | 72.24 | 83.56 | 87.58 | 89.18 | 89.76 | 90.23 | 90.56 | 90.64 | 90.83 | 90.91 | 87.55 |
| WAAL | 74.07 | 85.76 | 88.16 | 89.23 | 89.68 | 89.92 | 90.13 | 90.29 | 90.41 | 90.42 | 87.81 |
| **Proposed** | **78.22** | **87.37** | **89.03** | **90.11** | **90.60** | **90.72** | **90.91** | **91.26** | **91.43** | **91.74** | **89.14** |

Table 12: Results on imbalanced FashionMNIST.

