# OpenReview forum: "Distribution Aware Active Learning via Gaussian Mixtures"
_ICLR.cc/2024/Conference — Submitted to ICLR 2024_

### Official Review · Reviewer_aVJe · 2023-10-29

**Soundness:** 2 fair
**Presentation:** 3 good
**Contribution:** 2 fair
**Rating:** 3
**Confidence:** 4

**Summary:**

This paper presents a robust distribution-aware learning and sample selection strategy that employs Gaussian mixture model (GMM) to effectively encapsulate both labeled and unlabeled sets for AL.
A regularization method and an informativeness metric are further proposed to detect overfitted areas. Experiments on mulltiple datasets including balanced and imbalanced datasets demonstrate the validity of the proposed method.

**Strengths:**

(1) The solution of applying distribution alignment for active learning is valid.
(2) Multiple datasets including balanced and imbalanced datasets are evaluated to show the advantages of the proposed method.

**Weaknesses:**

(1)	It seems that the paper pose a relatively strong assumption that the data follows Gaussian mixture model (GMM) and derive the statistics from GMM. However, the author fails to show that GMM is a good model to approximate the data distribution in active learning. Actually, because of the nature of active learning, there are frequently samples from new classes that does not follow the estimated GMM model. In that case, the estimation based on GMM will not be accurate.
(2)	The novelty of the paper is limited as the distribution alignment has been widely studied in the previous work:
(i)	Zhang et al “Distribution Alignment: A Unified Framework for Long-tail Visual Recognition”. CVPR 2021
(ii)	“Agreement-Discrepancy-Selection: Active Learning with Progressive Distribution Alignment”, AAAI 2021
The reference (ii) also mentioned distribution alignment with adversarial learning but under a more general setting without GMM assumption. Thus, it seems this paper is talking about an existing method under some special conditions.

**Questions:**

As metioned in the weakness, the fundament problem of this paper is that it is merely a special case of a solved general problem, which makes the novelty of the paper very limited. Without detailed discussion of the difference from the existing work as listed, the novelty of the paper does not stay on a safe ground.

Moreover, the author needs to show why the GMM model is a good approximation of the data distribution in active learning as there can be many outliers and unknown classes. How about if the data is a long-tail distribution. The assumption of the paper is a bit too strong.

Last but not least, the paper did not any new insights of this field. The results are merely a small extension and special cases of existing approaches.

---

> ### Author Response · Authors · 2023-11-17
> **Response to Reviewer aVJe (Part 1)**
>
> We appreciate the reviewer for the time and efforts, and providing helpful comments and valuable feedback. Below, we provide responses to the reviewer's comments.
>
> ### **Is GMM a good model in real world AL settings?**
> As the reviewer correctly points out, in early learning cycles of AL, the labeled set ($X_L$) does not generally align well with the true distribution of the entire dataset. We agree that a single GMM would not give a good estimate here. However, note that in our approach we utilize two different GMMs $\psi_L$ and $\psi_{UL}$ for $X_L$ and $X_{UL}$; we fit new GMMs $\psi_L$, $\psi_{UL}$ at every AL cycles to reflect changes in $X_L$ and $X_{UL}$ such that new $\psi_L$ can encompass samples from new classes. This is how our method can handle samples from outliers and new classes. Related to this comment, we clarify that our intention is not to estimate the true distribution using $\psi_L$ alone, but to detect the discrepancy between $X_L$ and $X_{UL}$ using GMMs $\psi_L$ and $\psi_{UL}$. Our goal is then to “reduce this discrepancy” via distributional alignment in Section 3.4 and our informativeness-based data selection in Section 3.5.
>
> Moreover, our work does not assume special cases that completely exclude outliers or new classes, yet we achieve high performance in general AL settings adopted in previous AL works. To verify this further, we conducted additional experiments in two setups the reviewer suggested: 1) **unseen classes setting** where the initial labeled dataset only includes 70% of the entire label set, 2) **long-tail imbalance setting** that follows the common setup of [1,2] The results are provided in the tables below. It can be seen that our approach still outperforms other AL methods under these challenging settings. This further confirms the applicability and advantage of our method that uses two GMMs (for insights on why our method is more reliable, see later).
>
> - **Table 1**: Accuracy(%) in unseen classes setting on CIFAR10
>
> | # of labeled samples | 1K    | 2K    | 3K    | 4K    | 5K    | 6K    | 7K    | 8K    | 9K    | 10K   |
> |----------------------|-------|-------|-------|-------|-------|-------|-------|-------|-------|-------|
> | Random               | 39.19 | 65.69 | 74.62 | 78.46 | 80.77 | 82.55 | 83.54 | 85.50 | 86.02 | 86.30 |
> | TOD                  | 40.83 | 68.73 | 76.91 | 80.43 | 83.47 | 84.79 | 87.03 | 88.18 | 88.82 | 89.07 |
> | ALFA-Mix             | 39.19 | 67.86 | 76.36 | 79.92 | 83.29 | 84.83 | 87.23 | 88.52 | 89.03 | 89.31 |
> | TA-VAAL              | 39.19 | 69.01 | 77.81 | 80.03 | 82.94 | 84.56 | 86.11 | 87.64 | 88.03 | 88.44 |
> | DAAL                 | **42.95** | **72.51** | **79.37** | **82.47** | **84.58** | **85.98** | **88.07** | **89.36** | **89.59** | **90.04** |
>
> - **Table 2**: Accuracy(%) in long-tailed imbalance setting on CIFAR10 (imbalance ratio = 0.1)
>
> | # of labeled samples | 1K    | 2K    | 3K    | 4K    | 5K    | 6K    | 7K    | 8K    | 9K    | 10K   |
> |----------------------|-------|-------|-------|-------|-------|-------|-------|-------|-------|-------|
> | Random               | 45.59 | 57.33 | 68.42 | 71.36 | 76.07 | 76.86 | 78.04 | 79.93 | 81.64 | 82.37 |
> | CBAL [2]             | 45.59 | 61.32 | 71.67 | 74.93 | 78.09 | 79.20  | 81.87 | 82.36 | 83.12 | 85.51 |
> | DAAL                 | **50.92** | **71.87** | **77.98** | **81.47** | **82.48** | **85.34** | **86.53** | **87.09** | **87.74** | **88.53** |
>
> These results are now included in Sections A.4, A.5 of Appendix. Additionally, in Section A.7 of Appendix, we include experimental results on a different dataset (i.e., TinyImageNet) and backbone (i.e., ResNet-50).
>
> In summary, the use of two GMMs as suggested provide good estimates for data distribution in realistic active learning settings, as empirical results indicate.
>
> [1] Cui et al., "Class-balanced loss based on effective number of samples." CVPR 2019.
> [2] Bengar et al., “Class-balanced active learning for image classification”, WACV 2022.

---

> ### Author Response · Authors · 2023-11-17
> **Response to Reviewer aVJe (Part 2)**
>
> ### **Rationale behind the choice of GMM as an approximation model**
> Compared to other clustering algorithms such as K-means clustering, agglomerative clustering, and DBSCAN, GMM has the advantage of considering the desired number of modes along with the variance; it allows GMM to capture complex patterns in data with non-linear or non-spherical cluster shapes. Additionally, GMM is effective when clusters have overlapping boundaries due to its ability to allow for soft assignment, which takes into account ambiguity or uncertainty. Due to these advantages, GMM has been proven to be effective for the approximation of general real-world data (e.g., ImageNet, Cityscapes) in existing works [3,4]. Adopting GMM thus enables us to directly and accurately calculate the latent distribution as probabilistic models, providing a quicker and more reliable acquisition of distributional information compared to previous distributional alignment based AL approaches [5, 6] that indirectly infer the latent distribution by relying on pairwise distances or the output of additional modules.
>
> [3] Lee et al., "Meta-GMVAE: Mixture of Gaussian VAE for Unsupervised Meta-Learning." ICLR 2020.
> [4] Liang et al., “GMMSeg: Gaussian Mixture based Generative Semantic Segmentation Models”, NeurIPS 2022.
> [5] Fu et al., "Agreement-Discrepancy-Selection: Active Learning with Progressive Distribution Alignment", AAAI 2021.
> [6] Shui et al., "Deep Active Learning: Unified and Principled Method for Query and Training", AISTATS 2020.
>
> ### **Insights into the advantage of our method in realistic settings (e.g., unseen classes & long-tail distribution)**
>
> 1) Proposed likelihood informativeness ($I_{Like}$ in equation 4) can be calculated without any modification of our algorithm, even when the number of categories in $X_L$ and $X_{UL}$ differs (i.e., the number of modes in two GMMs is different). This is because we compare the likelihood score computed from each GMM. In a realistic imbalanced data setting, the likelihood difference between two GMMs ($\psi_L$, $\psi_{UL}$) is large around minor/unseen classes (i.e., large likelihood from $\psi_{UL}$ & low likelihood from $\psi_L$). Thus, the proposed informativeness score $I_{Like}$ prioritizes samples from minor/unseen classes to be selected, effectively addressing the imbalance problem.
> 2) Our semi-supervised learning with GMM also works well in an imbalanced setting. This is because, even if there are no or few samples for certain classes in $X_L$, GMM can capture modes for all classes based on information from $X_{UL}$. Therefore, our GMM-aided semi-supervised learning does not suffer from data imbalance, but rather encourages distributional alignment with the aid of GMM statistics from $X_{UL}$.

---

> ### Author Response · Authors · 2023-11-17
> **Response to Reviewer aVJe (Part 3)**
>
> ### **Is novelty limited in light of prior works? ((i) CVPR 2021 (ii) AAAI 2021)**
>
> **High-level differences with distributional alignment (DA) methods:** With regards to Distributional Alignment (DA) methods including references (i) and (ii), note that **our contribution is not on DA itself but more on capturing distributional information via GMM, for both DA and informativeness metric design.** As illustrated in Fig. 5 of the main paper, the discriminator of traditional DA methods tend to overfit to $X_L$, thereby generating overconfident and unreliable predictions (i.e., low entropy). In contrast, we propose to feed GMM statistics to the discriminator to improve efficiency of DA and predicted informativeness. Aside from distributional alignment, other features of our proposed methods are distribution-aware regularization and distribution-aware data selection; they work synergistically such that the distribution of $X_L$ quickly takes after the distribution of $X_{UL}$, thereby contributing to the significant performance improvement. Specifically, the idea of **computing informativeness based on the difference between likelihoods** has not been considered before, which is a key contribution of our work that can be achieved by taking advantage of GMMs.
>
>
> **More detailed technical differences with (i) and (ii):** With regards to the ADS work (reference (ii)), our method is different in the following senses. Whereas ADS leverages the difference between outputs of two adversarial classifiers to approximate the latent distribution, we directly apply the EM algorithm to the latent space. Taking the example of using ResNet18 on CIFAR10, we believe that our approach (which optimizes GMM on 512-dimensional latent space) is a more plausible design than ADS (which compares 10-dimensional classifier outputs) to estimate and align the latent distribution. Also, whereas ADS adopts adversarial learning that minimizes or maximizes the discrepancy between two adversarial classifiers, our work is based on a more solid theoretical background that optimizes the 1-lipschitz discriminator to minimize the Wasserstein distance between $X_L$  and $X_{UL}$. Furthermore, we enhanced the efficiency of distributional alignment by feeding the relative positional information (likelihood, posteriors) of training samples to the discriminator. We wanted to provide comparative results as well. However, note that reference (i) requires prior information about the true category distribution, which is not available in AL except for a small $X_L$. **For ADS (the reference (ii)) we ran experiments on CIFAR10 and CIFAR100.** The table is provided below and A.6 of Appendix. It can be seen that our work outperforms ADS. Finally, we note that we have already compared our method with **other well-known DA-based AL methods**, TA-VAAL and WAAL, in our original manuscript and validated our superiority (please refer to Figures 9 and 10 in our manuscript).
>
> - **Table 3**: Accuracy(%) on CIFAR10
>
> | # of labeled samples | 1K    | 2K    | 3K    | 4K    | 5K    | 6K    | 7K    | 8K    | 9K    | 10K   |
> |----------------------|-------|-------|-------|-------|-------|-------|-------|-------|-------|-------|
> | Random               | 51.76 | 63.50  | 72.17 | 76.94 | 80.56 | 82.32 | 84.17 | 85.00    | 86.11 | 87.27 |
> | ADS                  | 56.92 | 70.18 | 77.75 | 81.78 | 83.46 | 85.83 | 87.76 | 88.74 | 89.55 | 90.30  |
> | DAAL                 | **58.98** | **74.15** | **79.21** | **83.42** | **85.63** | **86.75** | **88.09** | **89.39** | **90.01** | **90.87** |
>
> - **Table 4**: Accuracy(%) on CIFAR100
>
> | # of labeled samples | 2K    | 4K    | 6K    | 8K    | 10K   | 12K   | 14K   | 16K   | 18K   | 20K   |
> |----------------------|-------|-------|-------|-------|-------|-------|-------|-------|-------|-------|
> | Random               | 20.68 | 31.76 | 41.34 | 47.56 | 51.65 | 55.80 | 58.07 | 59.66 | 61.53 | 63.07 |
> | ADS                  | 23.40 | 35.25 | 43.05 | 49.29 | 53.37 | 57.32 | 60.84 | 62.77 | 63.74 | 65.54 |
> | DAAL                 | **28.42** | **41.82** | **51.14** | **56.13** | **59.37** | **61.94** | **63.89** | **66.14** | **67.38** | **68.85** |
>
>
> *In summary, our method is very different from existing DA methods and also gives superior results.*
>
> (i) Zhang et al., “Distribution Alignment: A Unified Framework for Long-tail Visual Recognition”, CVPR 2021.
> (ii) Fu et al., "Agreement-Discrepancy-Selection: Active Learning with Progressive Distribution Alignment”, AAAI 2021.

---

> ### Author Response · Authors · 2023-11-17
> **Response to Reviewer aVJe (Part 4)**
>
> ### **Further Comment on the novelty of the proposed method**
>
> The novelty of our work lies in the **incorporation of GMM into training and selection methods that aim to minimize the discrepancy between $X_L$ and $X_{UL}$,** thereby mitigating the risk of overfitting to small $X_L$. Particularly, our approach involves the **estimation of latent distributions of $X_L$ and $X_{UL}$ using GMM**, a novel feature not observed in existing AL works. GMM is an useful tool for active learning since it can estimate the original data distribution even with the absence of certain categories in $X_L$, by leveraging the rich data in $X_{UL}$. Our proposed GMM-aided methods pose distinctive contributions that differ from prior works, which either train the task model using only $X_L$ or indirectly estimate the latent distribution without using probabilistic models like GMM. **Also, as demonstrated in Table 1 in the main paper, our GMM-aided components can be easily integrated into existing works, potentially enhancing performance.**
>
>
> Again, we appreciate the reviewer for the time and effort. Your concerns are clear, and we feel we have successfully addressed all the issues you raised. Please let us know in case there are remaining questions/concerns; we hope to be able to have an opportunity to further answer them.

---

> ### Comment · Reviewer_aVJe · 2023-11-22
> **Reviewer's comments**
>
> Thanks for the author's reply and comments.
>
> I have read through the response from the author and other reviewers' comments.
>
> I still have the fundamental questions that for active learning, due to the open-set annotations, using GMM for modelling is too restrictive as it imposed strong assumptions to the data. It is difficult to model other types of distributions. Namely one of my main concerns is from the fundamental math model.
>
> Moreover, regarding distribution alignment, I feel the current manuscript is merely applying the existing technique and applied it to the restrictive model (GMM) by ignoring other more common cases. The situation that discussed here is a special case of a more common problem which results in small novelty and limited scope.
>
> Therefore, based on the restrictive math modeling and limited novelty and scope as well as overlap with previous work, I maintain my review score.

---

> ### Author Response · Authors · 2023-11-22
> **Response to Reviewer aVJe**
>
> We appreciate your response. We would like to make the following points clearer:
>
>
>
> ### **`Generality/applicability of our work`**
>
>  We respectfully reiterate that **our algorithm is applicable to general AL settings**, and the **use of GMM does not limit the applicability** of our work. We also do not assume any special cases. To support these, we have provided comprehensive responses, conducted extensive experiments under the same/fair setting as other AL baselines in the manuscript, and showed general superiority in various datasets and AL scenarios **(balanced, imbalanced, long-tail, new classes)**, including the ones that the reviewer requested.
>
>
>
> ### **`Open-set annotation`**
>
>  In the original review, we believe that the reviewer didn’t mention the open-set annotation setting but instead suggested the “new classes” and “long-tailed” setup. This is why we have focused on the experiments mentioned above, and have explained why our scheme still works well with clear insights.
>
>  **Open-set annotation is a recently proposed new AL setting [CVPR’22] that deviates from the typical AL setting**, and there is still limited literature on it. It is assumed that the human annotator is not able to label certain unknown classes that are irrelevant to the main task. **Open-set annotation and conventional AL settings target different applications with different focuses**: For example, as provided in [CVPR’22], during AL for sports image classification, a large portion of images in the unlabeled set could be irrelevant to the main task (e.g., house, dogs). Open-set annotation setting will become important here. **However, the conventional AL setting (w/o open-set annotations) is still important in many cases**: In healthcare applications such as disease prediction, which is a primary application of AL, all images in the unlabeled set are directly related to the main task (e.g., X-ray image from the patient), which requires us to consider the conventional AL setup.
>
>  To handle the new open-set setting, an orthogonal approach to existing AL strategies is required: As we can see from [CVPR’22], all existing AL methods do not work well in this setting. Although this setup has not been actively studied, and unexplored, we can potentially address the challenge by orthogonally introducing an OOD sample detector as in [CVPR’22] to exclude OOD samples from the unlabeled set, and then apply our GMM-based algorithm.
>
>  [CVPR’22] Active Learning for Open-set Annotation
>
>
> ### **`Novelty`**
>
>  We would like to highlight that **the key philosophy of our work does not lie in distributional alignment (DA)**. DA (in Section 3.4) is just one of our components, and our contribution spans from GMM modeling (in Section 3.2) to regularization (in Section 3.3) and informativeness metrics (in Section 3.5), encompassing the entire AL scheme. **The core of this work is the use of GMM to better capture the dataset distribution and build a unified framework** for "distribution-aware AL" that facilitates effective regularization, effective DA-based learning, and effective informativeness-based sampling. We successfully integrate GMM into both the training and selection stages, aiming to identify and mitigate the discrepancies between the labeled and unlabeled sets.
>
>
>  During experiments, we also showed that **GMM can enhance the performance of various existing DA methods** and provided related analyses in Figure 5 of the manuscript and Section A.6 of the Appendix.  Moreover, as illustrated in Figure 12, **each proposed component exhibits a progressive enhancement** in performance. Furthermore, our compatibility analysis in Table 1 demonstrates that **our components can be easily plugged into the existing methods, leading to notable performance improvements**.
>
> Importantly, our work introduces a mathematical modeling of datasets employing GMM to enhance AL performance effectively.

---

### Official Review · Reviewer_wZHx · 2023-10-31

**Soundness:** 2 fair
**Presentation:** 3 good
**Contribution:** 2 fair
**Rating:** 6
**Confidence:** 5

**Summary:**

This paper introduces Gaussian Mixtures of labeled and unlabeled dataset to facilitate the model training, regularization, sample selection for active learning.

**Strengths:**

1. the improvement in performance over baselines is significant.
2. well-organised and easy to follow and understand, though there are many componenets in the proposed method.
3. The idea is original and it uses classific GMM to facilitate model training, as well as for high-quality and diverse sample selection for active learning.

**Weaknesses:**

1. The main concern would be the computation burden it introduces. With a batch, within each optimization iteration, there are 10 EM runs for both labelled and unlabelled samples. So it would be very time-consuming, even though there are some tricks used to reduce it. So both theoretical and experimental analysis about time complexity would be expected.

2. Some details about loss functions are missing to better understand the training. For example, there is a trade-off parameter $\alpha$ to balance cross-entropy loss and regularization loss. The value for this constant was not given. Besides, there is another loss for adversarial learning. How to balance with this function is not clear to me.

3. Some intuitive explanation to demonstrate the superiority would be great. For example, with each active learning cycle, for each class, how the selected samples from the proposed method are different from the ones from baselines? Or use the selected samples to verify the claim of 'high quality and diverse'.

**Questions:**

see above weakness points.

---

> ### Author Response · Authors · 2023-11-18
> **Response to Reviewer wZHx (Part 1)**
>
> We appreciate the reviewer's valuable comments on our paper. Our answers to the reviewer's comments are given below:
>
> ### **GMM time complexity**
> We appreciate the reviewer’s concern regarding the complexity issue. In our work, we compute the GMM in both the “supervised learning stage” and the “semi-supervised learning stage”. GMM calculations impose an increased computational burden of approximately **5.66%** and **15.19%** in each respective stage. For detailed explanations, please refer to the information provided below with our Algorithm in Section A.2 in Appendix.
>
> **Supervised learning stage.** We calculate the GMM parameters $\psi_L$ for the labeled set ($X_L$) once per minibatch (we do not compute the GMM for $X_{UL}$ at this stage). As we have label information, we can directly compute $\psi_s≔(\pi_s , \mu_s, \sigma_s^2)$ without multiple iterations of the EM algorithm. Detailed analysis on the operational time is provided below.
>
> - **Table 1**: Analysis of the required time to process one batch in the **supervised learning**
> | Is GMM-related? | Algorithm procedure| Time required |
> |----------------------|-------|-------|
> | x | Draw a mini batch ($B$) from  the dataloader | 33.1505 msec.  |
> | o | Draw a i.i.d. subset ($X_s$) from the dataloader | 4.5133 msec. |
> | o | Pass $X_s$ through the feature extractor $f_\theta$ to get $f_\theta(X_s)$ | 6.1559 msec. |
> | o | Caclulate GMM $\psi_s$ from $f_\theta(X_s)$ | 0.0470 msec. |
> | x | Compute $Loss_{CE}$, $Loss_{GMM}$ using minibatch (B) and GMM $\psi_s$ | 3.7744 msec. |
> | x | Update model by minimizing $Loss_{CE}$, $Loss_{GMM}$ | 141.7344 msec. |
>
> **Semi-supervised learning stage.**  This stage consists of a total of 10,000 iterations (batches), and GMMs $(\psi_L,\psi_{UL})$ are computed at every 500 iterations for semi-supervised learning. Similiarly with the above, $\psi_L$ can be directly computed using the labels, and only $\psi_{UL}$ is computed via an iterative EM algorithm. The computation of GMM accounts for 15.19% of the entire time required for semi-supervised learning, and a detailed time analysis is provided below.
>
> - **Table 2**: Analysis of the required time to process one the **semi-supervised learning**
> | Algorithm procedure | Time required |
> |-------|-------|
> | Time required for 500 iterations (including GMM calculation) | 144.9939 sec.  |
> | Time required for GMM calculation | 22.0256 sec.|
> | Ratio of time required for GMM calculation during semi-supervised learning | 15.1906 % |
>
> These additional delays are the costs for achieving significant improvements in AL performance. Based on the reviewer’s comments, we have now included these results in Section A.8 of the Appendix.
>
> **Theoretical analysis on GMM time complexity**.
> Basically, the time complexity for the computation of vanilla GMM is **$O(NKID^2)$**, where $N, K, I, D$ represent the number of data, number of Gaussian modes, number of EM iterations, and the dimensionality of the data. However, we reduced the complexity to **$O(NKID)$ by making the isotropic Gaussian assumption**, which helps accelerate GMM computation. Moreover, as evident from the above experimental analysis, the time required for GMM computation is not significantly demanding compared to the time spent on forward/backward propagation for model update.

---

> ### Author Response · Authors · 2023-11-18
> **Response to Reviewer wZHx (Part 2)**
>
> ### **Values of hyperparameters that balance multiple losses**
> We appreciate the constructive comments. The value of $\alpha$ is fixed to be 1e-4, and details on hyperparameters including $\alpha$ can be seen in Section A.1 of Appendix. In accordance with the reviewer’s comments, we compared the performance by varying the value of $\alpha$ from 0.00001 to 0.001 and reported the results below. When $\alpha$ is too large, distributional regularization on the latent space compromises the performance of the main task. On the other hand, when $\alpha$ is too small, the effect of regularization and performance improvement diminishes. These results are now included in Section A.9 of Appendix.
>
> - **Table 3**: Accuracy(%) on CIFAR10
>
> | # of labeled samples | 1K    | 2K    | 3K    | 4K    | 5K    | 6K    | 7K    | 8K    | 9K    | 10K   |
> |----------------------|-------|-------|-------|-------|-------|-------|-------|-------|-------|-------|
> | $\alpha=0.001$               | 59.24 | 73.96 | 77.81 | 82.81 | 85.22 | 85.88 | 87.60 |	88.13 | 88.96 |	89.53 |
> | $\alpha=0.0005$             | 59.21 | 74.07 | 78.98 |	82.73 | 85.40 |	86.67 | 87.57 |	89.04 | 89.57 |	90.19 |
> | $\alpha=0.0001$             | **60.64** | **74.15** | **79.21** |	**83.42** | **85.63** | **86.75** | **88.09** | **89.39** | **90.01** |	**90.87** |
> | $\alpha=0.00005$           | 58.98 | 73.63 | 77.45 | 82.16 | 84.76 |	85.88 | 87.43 |	88.95 | 89.63 |	90.38 |
> | $\alpha=0.00001$           | 59.58 | 73.41 | 76.52 |	81.78 | 84.25 |	86.22 | 87.32 |	88.83 | 89.44 | 90.01 |
>
> As for the loss functions during adversarial training, we use two losses: cross-entropy loss $L_{CE}$ and discriminator loss $D_\phi (x_{ul}) - D_\phi (x_l)$. Without a special constant weight to balance them, we added two losses in a 1:1 ratio and conducted the adversarial semi-supervised learning. We have made this clearer in Section A.9 of Appendix.
>
>
> ### **Intuitive explanation on the superiority of our method over other baselines.**
>
> In response to the reviewer’s comment, in **Fig. 20 of Section A.10 in Appendix**, we illustrate the candidate set selected by various AL methods under imbalanced CIFAR10. From the t-SNE visualization of candidate sets, we can intuitively compare the properties of various AL methods. For instance, Snapshot_AL [1] that calculates uncertainty using the output of the classifier exhibits redundancy by selecting multiple samples from narrow regions. Coreset [2] that considers only diversity can select diverse samples in broad areas but it lacks the ability to specify certain informative regions. CoreGCN [3] that favors samples expected to be from unlabeled set, is observed to select samples from dense areas of $X_{L}$, as well as outlier samples that hardly represent $X_{UL}$. In contrast, the candidate set from the proposed method demonstrates desired properties: **1) it is diverse enough to encompass a broad area** while **2) represents the distribution of $X_{UL}$ well. Our method compares GMMs of $X_L, X_{UL}$ to select samples that are unfamiliar in $X_L$, but representative of $X_{UL}$**; especially in the imbalanced setting, this comparison leads to select minor classes that are rare in $X_L$, thereby contributing to the significant performance gain as seen in Fig.10 of the main paper.
>
> Additionally, we reported the class count entropy of $X_L$ under an imbalanced setting in Figure 11 (b) of the original manuscript (i.e., higher entropy corresponds to diverse $X_L$). The results show that the proposed method actively selects minor classes, exhibiting the highest entropy.
>
> [1] Jung et al., “A Simple Yet Powerful Deep Active Learning With Snapshots Ensembles”, ICLR 2023.
> [2] Sener et al., “Active Learning for Convolutional Neural Networks: A Core-Set Approach”, ICLR 2018.
> [3] Caramalau et al., “Sequential graph convolutional network for active learning”, CVPR 2021.
>
> We again appreciate the reviewer for providing helpful comments. We would be grateful if you could let us know if there are any other comments/questions.

---

> > ### Comment · Reviewer_wZHx · 2023-11-23
> >
> > I have carefully examined the response from the authors, and most of my concerns have been addressed.

---

### Official Review · Reviewer_ZH7N · 2023-11-02

**Soundness:** 3 good
**Presentation:** 3 good
**Contribution:** 3 good
**Rating:** 8
**Confidence:** 4

**Summary:**

This paper introduces a new active learning strategy that addresses the issue of mismatched distributions between labeled and unlabeled samples by incorporating a Gaussian Mixture Model (GMM). The strategy combines various informativeness metrics for sample selection. Tests on different datasets show that this GMM-based method performs better than existing approaches and can be combined with other active learning methods to enhance results further.

**Strengths:**

This paper introduces a new active learning strategy by incorporating a Gaussian Mixture Model (GMM). This strategy aims to address the issue of mismatched distributions between labeled and unlabeled samples.

This GMM-based method performs better than existing approaches on several datasets.

**Weaknesses:**

If I understand correctly, in Section 3.2, two GMMs are fitted for the labeled and unlabeled data, respectively. Is this correct? Equation (2) provides a means to compare two Gaussian distributions, but it is not immediately clear how this extends to the comparison of two GMMs, and whether the weight of each Gaussian component in the mix is considered in this comparison.

Regarding Equation (3), the purpose of the optimization seems to be to minimize the disparity between the unlabeled data, $X_{UL}$, and the labeled data, $X_{L}$, within the embedding space. If the model is trained effectively, it appears that $X_{UL}$ and $X_{L}$ would become indistinguishable in the embedding space. It raises the question of whether this is the intended outcome, as it would seem important for the embedding space to retain the information that differentiates $X_{UL}$ from $X_{L}$.

The combination method outlined in Equation (5) is not entirely convincing as the best approach. An alternative sampling strategy might involve drawing from each of the three separate rankings and then using the highest-ranked samples from each ranking as the data set for annotation. It would also be insightful to see the results of an ablation study where each component of Equation (5) is used independently as the final selection criterion.

In Equation (5), it seems that only $I_{Ent}$ leverages the labels from previous iterations. The other two components are influenced by $X_{UL}$ and $X_{L}$ but are label-independent. It may be beneficial for the authors to make this point clearer in their writing.

**Questions:**

1. Are two GMMs fitted for the labeled and unlabeled data, respectively?
2. How does Equation (2) extend to the comparison of two GMMs?
3. What is the rationale behind Equation (3)?
4. Why Equation (5) is a good way to combine the information?
5. In Equation (5), is $I_{Ent}$ the only component that leverages the labels from previous iterations, but the other components label-independent?

---

> ### Author Response · Authors · 2023-11-18
> **Response to Reviewer ZH7N (Part 1)**
>
> We appreciate the reviewer for taking a positive stance and providing helpful comments on unclear aspects and constructive suggestions. Below, we tried to carefully address the concerns raised by the reviewer.
>
> ### **Are two GMMs fitted for $X_L$ and $X_{UL}$, respectively?**
>
> As the reviewer correctly pointed out, we need to obtain GMM parameters ($\psi := \pi, \mu, \sigma^2$) in latent space for the label set ($X_L$) and the unlabeled set ($X_{UL}$), respectively. For $X_{UL}$, a GMM $\psi_{UL}$ is fitted via EM algorithm. However, for $X_L$, since the label information is given, GMM parameters $\psi_{L}$ for each mode (i.e., class) can be efficiently obtained by directly computing mean and variance for each class without performing iterative EM algorithm.
>
> ### **Detailed explanation regarding Equation (2).**
>
> As mentioned above, since label information is available when obtaining a GMM $\psi_{L}$, each Gaussian mode of the fitted GMM can be matched to each category (class) in the labeled data. However, when a GMM $\psi_{UL}$ is fitted to the unlabeled samples via EM algorithm, the order of clusters can be scrambled since label information is unknown for the unlabeled samples. In this context, the Bhattacharyya distance in Eq. (2) is utilized to align each mode in the unlabeled GMM $\psi_{UL}$ with the corresponding category (mode) in the labeled GMM $\psi_{L}$, through Hungarian matching problem.
> **Note that the intention behind using Eq. (2) is not to compare or calculate the distance between GMMs, but rather to compute the distance between two Gaussian distributions from each GMM for $X_L$ and $X_{UL}$ for alignment**. Therefore, the mixing coefficient ($\pi_k$) does not play a meaningful role in this context. Furthermore, the mixing coefficient ($\pi_K$) in GMMs reflects the occurrence frequency of samples for each category. Therefore, in real-world scenarios such as imbalance settings where the class distributions of labeled and unlabeled sets are different, considering this coefficient ($\pi_k$) to align the Gaussian modalities from two different GMMs can hamper correct matching. For this reason, we solve the matching problem through equation (2) without considering the coefficient ($\pi_k$).
>
> ### **Rationale behind Equation (3)**
> As the reviewer mentioned, the goal of Equation (3) is to perform adversarial training based on GMM information to reduce the distribution discrepancy between $X_L$ and $X_{UL}$ in latent space, and it offers important advantages in AL. This is because models are susceptible to overfitting due to the limited size of $X_L$ in the early stages of AL (Distinct difference between the distributions of $X_L$ and $X_{UL}$ can be seen in Fig. 1 of the main paper).
> Considering the nature of AL, we believe it is almost impossible for Equation (3) to align $X_L$ with $X_{UL}$ to be indistinguishable in the embedding space during challenging early AL cycles. As pointed out by the reviewer, there might be a moment when it is possible for Equation (3) to align $X_L$ and $X_{UL}$ when the size of $X_L$ grows enough. However, this implies that training the model using only $X_L$ can achieve performance similar to using $X_L$ + $X_{UL}$, overshadowing the need to apply active learning.

---

> ### Author Response · Authors · 2023-11-18
> **Response to Reviewer ZH7N (Part 2)**
>
> ### **Ablation study on Equation (5)**
> We thank the reviewer for the insightful comments. As can be seen in Fig. 8 (a) and (c) of the main text, two metrics ($I_{Like}$, $I_{Ent}$) exhibit strong redundancy in their candidate set. Therefore, separately considering each metric ($I_{Like}$, $I_{Dis}$, $I_{Ent}$) can lead to redundant and overlapping selections. However, when considering a combined indicator as in our method ($I_{Total}$), one can select diverse yet informative samples as shown in Fig. 8 (d). In response to the reviewer’s comment, we conducted an additional experiment in which data selection is performed based on separate rankings for each informativeness score ($I_{Like}$, $I_{Dis}$, $I_{Ent}$). Through the table below, we confirmed that our integrated indicator achieves higher performance than using each independent indicator.
>
> - **Table 1**: Comparison with the sampling strategy based on separate rankings for each informativeness score on CIFAR10
>
> | # of labeled samples | 1K    | 2K    | 3K    | 4K    | 5K    | 6K    | 7K    | 8K    | 9K    | 10K   |
> |----------------------|-------|-------|-------|-------|-------|-------|-------|-------|-------|-------|
> | Random               | 51.76 | 63.50  | 72.17 | 76.94 | 80.56 | 82.32 | 84.17 | 85.00    | 86.11 | 87.27 |
> | Separate rankings for ($I_{Like}$, $I_{Dis}$, $I_{Ent}$)           | 58.98| 71.79| 78.79| 82.68| 84.96| 86.31| 87.69| 88.63| 89.37| 90.08 |
> | DAAL                 | **58.98** | **74.15** | **79.21** | **83.42** | **85.63** | **86.75** | **88.09** | **89.39** | **90.01** | **90.87** |
>
> ### **Does $I_{Ent}$ leverage label information?**
> We would like to clarify that after the training is finished, we measure the three informativeness metrics ($I_{Like}$, $I_{Dis}$, $I_{Ent}$) on the unlabeled set $X_{UL}$ to select unlabeled samples with high informativeness. Specifically, we only utilize label information of labeled set $X_L$ when training the model in the supervised and semi-supervised learning stages. In the sample selection stage, none of the three informativeness metrics utilize label information: The goal is to select the best samples from the "unlabeled set" to be added in the labeled set. Regarding $I_{Ent}$, we compute Shannon entropy to compute $I_{Ent}$ as: $I_{Ent}(x_{ul}) = - \sum_{k=1}^{K} p(y=k|x_{ul}) log(p(y=k|x_{ul}))$, where $p(y=k|x_{ul})$ is the classifier output (not label information) after the softmax function for class $k$ given the unlabeled sample $x_{ul}$. As for remaining metrics $I_{Like}$ and $I_{Dis}$, the two GMMs ($\psi_{L}$, $\psi_{UL}$) for $X_{L}$ and $X_{UL}$ are utilized to compute $I_{Like}$, and the discriminator trained on $X_{L}$ and $X_{UL}$ is utilized to $I_{Dis}$. Based on the reviewer’s comment, we have made these points clearer in Section 3.5 of the revised manuscript.
>
> Again, we appreciate the reviewer for the time and positive position. Your comments made us think carefully, and we feel we have successfully addressed and clarified all the comments you raised. Please let us know in case there are remaining questions/concerns; we hope to be able to have an opportunity to further answer them.

---

### Meta-Review · Area_Chair_XcdU · 2023-12-09

**Metareview:**

The paper introduces an novel active learning strategy using Gaussian Mixture Models (GMM) to address the distributional discrepancies between labeled and unlabeled datasets. This approach aims to reduce overfitting and enhance the diversity of feature representation by employing a combination of regularization, distributional alignment, and a hybrid sample selection strategy.

The reviewers acknowledged the novel integration of GMM in AL, the extensive experiments across various datasets, and substantial theoretical contributions. While the majority of the reviewers are positive, there is a concern regarding the complexity and assumptions regarding the GMM's ability to model data distributions in AL, particularly in scenarios involving new classes or outliers. This issue unfortunately was not fully resolved even after the discussion phase. The authors are encouraged to consider these points in future work, e.g., to adjust the positioning of the paper accordingly to further clarify and enhance the applicability of their work.

**Justification For Why Not Higher Score:**

The paper does not receive a higher score primarily due to the concerns about its foundational assumptions and the novelty of its contributions. The reliance on GMM to model data distributions in active learning scenarios, especially those with new or outlier classes, is seen as a potential limitation. This modeling choice raises questions about the applicability and generality of the proposed method in diverse real-world scenarios.

Despite the novel integration of GMM, the core concept of distribution alignment has been previously explored in the literature, which limits the perceived novelty of this work. These aspects suggest that while the paper is a strong contribution to AL, it may not be sufficiently novel to warrant a higher score.

**Justification For Why Not Lower Score:**

The novel use of GMM in this context and the development of new regularization and sample selection strategies are well recognized by the reviewers. The extensive experiments and results showcase significant performance improvements over existing methods. The authors' engagement in the review discussions, particularly in addressing questions about the runtime complexity and providing additional empirical evidence, further supports the paper's standing.

---

### Decision · Program_Chairs · 2024-01-16

Reject